# A benchmark for prediction of transcriptomic responses to chemical perturbations across cell types

**Artur Szałata**[1*]**, Andrew Benz**[2*]**, Robrecht Cannoodt**[3,4,5]**, Mauricio Cortes**[2]**, Jason Fong**[2]**,
**Sunil Kuppasani**[2]**, Richard Lieberman**[2]**, Tianyu Liu**[6]**, Javier A. Mas-Rosario**[2]**,
**Rico Meinl**[7]**, Jalil Nourisa**[8]**, Jared Tumiel**[7]**, Tin M. Tunjic**[9]**, Mengbo Wang**[10]**,
**Noah Weber**[11]**, Hongyu Zhao**[6]**, Benedict Anchang**[12]**, Fabian J. Theis**[1†]**,
**Malte D. Luecken**[1†]**, Daniel B. Burkhardt**[2†]

[1]Helmholtz Munich, [2]Cellarity, [3]Data Intuitive, [4]VIB Center for Inflammation Research,
[5]Ghent University, [6]Yale University, [7]Retro Biosciences, [8]Helmholtz Center Hereon,
[9]TU Vienna, [10]Purdue University, [11]Olden Labs, [12]NIH, [*,†]Equal contribution
`{artur.szalata, fabian.theis, malte.luecken}@helmholtz-munich.de;`
`{abenz, dburkhardt}@cellarity.com`

## Abstract

Single-cell transcriptomics has revolutionized our understanding of cellular hetero-
geneity and drug perturbation effects. However, its high cost and the vast chemical
space of potential drugs present barriers to experimentally characterizing the effect
of chemical perturbations in all the myriad cell types of the human body. To
overcome these limitations, several groups have proposed using machine learning
methods to directly predict the effect of chemical perturbations either across cell
contexts or chemical space. However, advances in this field have been hindered
by a lack of well-designed evaluation datasets and benchmarks. To drive innova-
tion in perturbation modeling, the Open Problems Perturbation Prediction (OP3)
benchmark introduces a framework for predicting the effects of small molecule per-
turbations on cell type-specific gene expression. OP3 leverages the Open Problems
in Single-cell Analysis benchmarking infrastructure and is enabled by a new single-
cell perturbation dataset, encompassing 146 compounds tested on human blood
cells. The benchmark includes diverse data representations, evaluation metrics,
and winning methods from our "Single-cell perturbation prediction: generaliz-
ing experimental interventions to unseen contexts" competition at NeurIPS 2023.
We envision that the OP3 benchmark and competition will drive innovation in
single-cell perturbation prediction by improving the accessibility, visibility, and
feasibility of this challenge, thereby promoting the impact of machine learning in
drug discovery.

## 1 Introduction

Examining gene expression in individual cells via single-cell RNA sequencing (scRNA-seq) provides
high-resolution insights into cellular behavior within healthy and diseased tissue. One emerging
application of single-cell technology is to profile cells under basal and perturbed states to characterize
the changes in cellular states associated with chemical treatments and to associate these changes with
healthy or pathological tissue phenotypes [1–5]. These technologies have the potential to transform

Submitted to the 38th Conference on Neural Information Processing Systems (NeurIPS 2024) Track on Datasets
and Benchmarks. Do not distribute.

how drugs are discovered and bring new therapies to patients with unmet clinical needs [6–8]. Instead of focusing on single molecular targets for drug discovery, it is possible to analyze how compounds influence gene expression to shift cells from diseased to healthy states. This approach holds promise for treating complex diseases where single-target methods have been less effective, as it addresses the interplay of multiple genes and pathways within the cell.

However, associating small molecules with changes in cell state is challenging. One approach is to brute-force screen compounds and measure the associated changes in gene expression, as has been done to discover drug candidates for heart valve disorders [9]. However, chemical space is vast. There are an estimated $10^{60}$ drug-like molecules [10]. Compounds can also have diverse impacts on gene expression across different tissues, cell types, and individuals. Moreover, scRNA-seq experiments are expensive and require highly-trained technicians to run. Hence, accurate prediction of the changes in gene expression induced by compounds across different chemical structures and biological contexts could provide immense time and cost savings.

Recently, machine learning methods to predict the impact on gene expression of small molecule perturbations directly from chemical structures have been proposed [11–14]. However, understanding such models' effectiveness is difficult due to a lack of independent evaluations and limited availability of benchmarking datasets [15]. Indeed, most existing datasets include only a single perturbation [16], a single donor, or are limited to homogeneous cancer cell lines [1, 17]. Although these studies represent important contributions to the field, a rigorous, standardized benchmark is needed to assess their performance in diverse cell types across a wide range of chemical perturbations.

Here, we introduce the Open Problems Perturbation Prediction (OP3) benchmark, which is the first standardized benchmark for predicting chemical perturbation effects across cell types. It includes a formalized task, an open-source benchmarking platform, and a new dataset profiling 146 chemical perturbations in human peripheral blood mononuclear cells (PBMCs) from three donors. We hosted a NeurIPS 2023 Competition using this benchmark, and used the learnings and proposed methods to improve the benchmark. OP3 provides a continuously updated, extensible benchmark for perturbation prediction, promoting translation of these methods to applied science.

## 2  Related work

This work builds on previous efforts to generate single-cell chemical perturbation datasets and evaluations performed alongside method development for perturbation prediction algorithms.

**Chemical perturbation datasets**   Recently, several large-scale datasets with drug perturbations have been published. The popular sci-Plex [17] dataset profiles 188 compounds in three cancer cell lines, and its recent sequel, the sci-Plex-GxE [18] dataset, profiled 22 drugs combinatorially in three cancer cell lines. While these datasets feature a large number of compounds, their use of cancer cell lines limits their applicability, as cancer cell lines have a number of significant deviations from human tissue. These datasets also use nuclei sequencing technologies which are less sensitive and have higher noise compared to whole-cell sequencing used in our study [19]. In addition, a recent pre-print introduced a scRNA-seq dataset of drug-perturbed human PBMCs [20], but its lack of replicates makes it difficult to disentangle technical and biological noise from the drug perturbation signal. Finally, a harmonized collection of public single-cell perturbation datasets was recently published, but most datasets contain only a single cell type and few perturbations with overlap across datasets, making them unsuitable for our benchmarking task [15].

**Perturbation prediction evaluation**   The task of predicting the transcriptomic effects of small molecule perturbations in single-cell data has been tackled by a few machine learning models [13, 14, 12, 21]. However, the evaluations of these models did not include drug perturbations on primary tissue, used evaluation methods that are biased toward natural transcriptional variation [22], and lacked assessments of stability across replicates and batches. No independent method evaluations exist to our knowledge, which is essential to fairly compare algorithm performance [23].

# 3 A living benchmark for perturbation prediction

To drive innovation in algorithm development for single-cell perturbation analysis, we set up the OP3 benchmark, including a formalized task definition, a fit-for-purpose benchmarking dataset, and computational infrastructure to support continuously-updated, community-driven benchmarking (**Figure 1a**). We outline these features below.

## 3.1 Task overview

Chemical perturbations induce cell type-specific gene expression changes by interacting with target proteins and altering cellular processes. For example, tamoxifen, a breast cancer drug, binds the estrogen receptor and inhibits cell growth, thereby acting selectively on cells expressing the estrogen receptor [24]. However, the lack of knowledge about mechanisms of action for most compounds hinders predicting their effects on specific cell types.

The goal of this task is to leverage data about chemical perturbations in some cell types to infer their impact on gene expression in other cell types. The data is a tensor with three axes: compounds, cell types, and genes. Each value in this tensor is a measurement of the impact on gene expression observed in a specific cell type under a specific chemical perturbation (**Section 3.3**). Models are provided with the changes in gene expression for all cell types for a subset of compounds. The remaining compounds comprise the test set. These compounds have their differential expression values masked for all genes for a subset of the cell types. The target of this task is to predict these masked differential expression values (**Figure 1b**).

## 3.2 Generating a single-cell perturbation benchmarking dataset

**Considerations for data set generation**   We identified the following properties of an ideal dataset for benchmarking small molecule perturbation prediction:

1. **Disease-relevance:** To reflect the downstream application to drug discovery, an ideal dataset ought to focus on a disease-relevant biological system.

2. **Balanced cellular heterogeneity:** Cell types must exhibit distinct perturbation responses but be similar enough that translating compounds' effects is tractable.

3. **Diverse perturbations:** The compounds should perturb a range of biochemical pathways.

4. **Replicates across multiple donors:** Capturing perturbation effects across multiple donors enables identifying effects that are preserved across diverse donors.

5. **Positive and negative controls**: Because of the high degree of technical and biological variability in gene expression measurements, positive and negative controls are essential to accurately estimate the variation attributable to perturbation effects.

6. **Open access & informed consent:** To ensure open access to benchmarking data collected from human donors, samples must be collected under IRB supervision. This ensures donors give informed consent for public sharing of any derived data.

**Dataset overview**   We generated a novel scRNA-seq dataset profiling 146 compounds in PBMCs to provide a high-quality reference benchmark dataset for single-cell perturbation prediction (**Figure 1c**). We also included multiome single-nucleus RNA and chromatin accessibility measurements at baseline to facilitate gene regulatory network inference. This effort represents, to date, the largest drug perturbation dataset on primary human tissue with donor replicates [15], and was specifically designed to satisfy all the criteria above. First, PBMCs comprise an important subset of the human immune system and play a key role in various pathologies, including cancer, autoimmune diseases, immunodeficiencies, and allergies. PBMCs also contain discrete cell types (including T-cells, B-cells, myeloid cells, and NK cells) that perform distinct biological functions while sharing key biological pathways, making perturbation prediction in PBMCs difficult yet tractable. The compounds in this dataset were selected to span a wide range of mechanisms of action. Additionally, two positive control

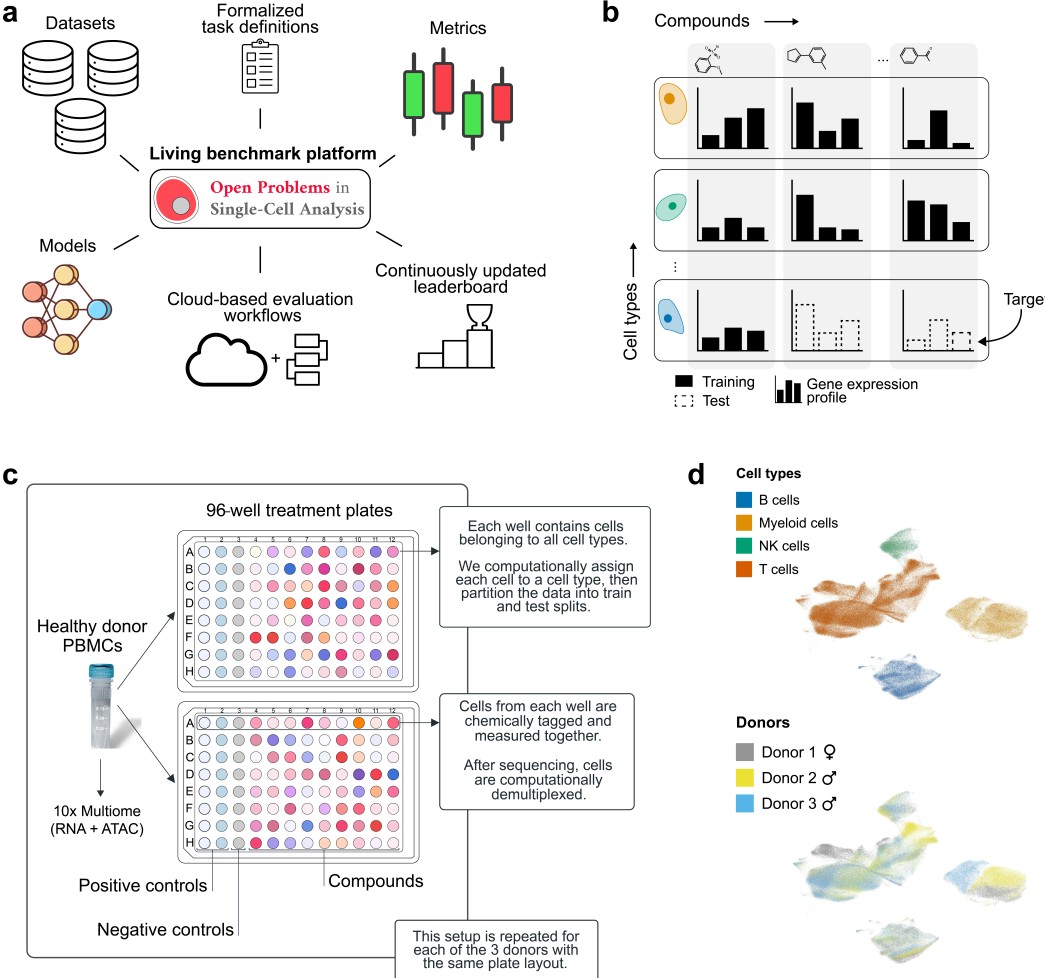

Figure 1: **Overview of the dataset.** **(a)** A overview of the Open Problems living benchmarking framework. **(b)** A graphical description of the perturbation prediction task. **(c)** The experimental setup for our benchmarking dataset. **(d)** UMAP representations of the resulting single-cell profiles colored by cell type (top) and donor (bottom).

compounds that were known to induce a strong transcriptional signature in PBMCs were included. Every perturbation was repeated in three healthy human donors, two male and one female. Finally, we performed this experiment using PBMCs that were commercially available with pre-obtained consent for public release.

**Data generation, processing and cell type annotation** PBMCs were cultured in six separate 96-well plates, two for each donor (**Figure 1c**). After the cells were treated with compounds for 24 hours, samples were collected, pooled to reduce batch effects and increase throughput, and sequenced. Sequencing reads were processed using the Cell Ranger pipeline [25], and a best-practice pipeline was followed to QC, normalize, reduce, and cluster the data [26]. We assigned each cluster to one of four cell type labels (B cell, T cell, NK cell, or myeloid cell) using established marker genes. **Figure 1d** shows the UMAP [27] visualization of the dataset with cell type and donor annotations.

The baseline multiome data (joint snRNA-seq/scATAC-seq) was processed by filtering out low-quality cells, along with both genes and chromatin accessibility features with low counts. Cells in this multiome data were then annotated based on marker gene expression in the same manner as the

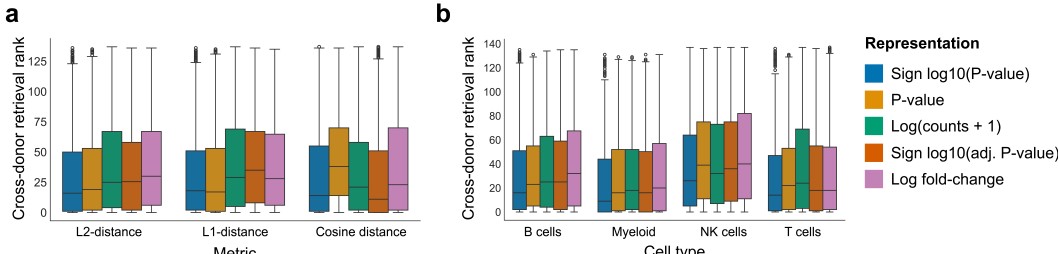

Figure 2: **Cross-donor retrieval analysis.** (**a**) For each pair of donors, for each compound in each cell type, the cross-donor retrieval rank was calculated using various distance metrics. The y-axis shows the retrieval rank (i.e., the rank of the same compound and cell type measurement in a different donor). The x-axis separates different retrieval distance metrics. Note that L1 distance is effectively a rescaled MAE, and L2 distance is effectively a rescaled RMSE. The hue differentiates box plots for different data representations according to the legend on the right. (**b**) We further examined the cross-donor retrieval rank per cell type using the L2-distance metric to ensure the results were consistent across cell types.

perturbational scRNA-seq data. For a detailed description of the experiment and analysis for both perturbational and baseline multiome data, please refer to **Appendix A**.

### 3.3 Representation of perturbation effects

In genomics, differential expression (DE) analysis is commonly used to identify how compounds affect gene activity in different cell types [26]. DE methods estimate perturbation effects by fitting generalized linear models to observed count data, explicitly accounting for biological and technical covariates. In this study, we performed DE analysis using the limma-voom framework [28], which provides estimates of effect size (e.g., log-fold change) and statistical significance while adjusting for variability associated with technical covariates.

Although using estimates of effect size or significance is standard in the genomics community, it is more common in machine learning benchmarks to directly predict a conditional distribution, such as the gene expression counts. To test whether the effect size (log-fold change), significance ($p$-values), or conditional counts are more suitable for benchmarking, we evaluated each of these representations using the replicates across donors in our dataset. We determined that an optimal representation would minimize the distance between observations of the same compound across donors, with lower median distance ranks indicating better identifiability of compounds across donors. We call this heuristic *cross-donor retrieval* (**Appendix C.1**).

We found that the measures of effect significance had better cross-donor retrieval (**Figure 2a** and **Appendix Figure 5**) than effect size or counts data, and this effect was consistent across cell types (**Figure 2b**). Based on these results, we decided on the following representation as a target for our benchmark: for a given compound $c$, cell type $t$, and gene $g$, let $p_{c,t,g}$ and $L_{c,t,g}$ be the $p$-value and log-fold change computed by `limma`, respectively. Then

$$\text{pert}_{c,t,g} = -\log_{10}(p_{c,t,g}) \times \text{sign}(L_{c,t,g}). \tag{1}$$

This representation captures both the direction and statistical significance of the perturbation effect on each gene. We do not claim that this representation is universally optimal for all tasks and analyses and note there are several challenges associated with DE analysis generally (**Appendix D**).

### 3.4 Evaluation metrics

We considered three metrics for evaluating model performance: mean row-wise root mean squared error (MRRMSE), mean absolute error (MAE), and cosine similarity. Mean row-wise indicates that we take a mean across predictions for compound-cell type pairs. Each of these metrics is related to the distance metrics used in the cross-donor retrieval task, e.g. MAE is effectively a rescaled

L1 distance, and MRRMSE is effectively a rescaled L2 distance. Using these relationships, we concluded that cosine similarity had the best stability across donors, followed by MRRMSE and MAE (**Figure 2a**). However, not all perturbations are expected to cause a change in gene expression, and cosine similarity would not penalize models that incorrectly predict low $p$-values in such cases, unlike MRRMSE and MAE. Hence, we primarily rely on MRRMSE for model evaluation, defined as:

$$\text{MRRMSE} = \frac{1}{R}\sum_{i=1}^{R}\left(\frac{1}{n}\sum_{j=1}^{n}(y_{ij}-\hat{y}_{ij})^2\right)^{1/2} \tag{2}$$

Where $R$ is the number of (cell type, compound) tuples, and $y_{ij}$ and $\hat{y}_{ij}$ are the actual and predicted values, respectively, and $n$ is the number of genes.

### 3.5 Control methods

Including control methods in each benchmarking task is one of the basic quality controls required by Open Problems not only to verify the integrity of the benchmarking workflow but to also normalize the metric outputs. In this benchmark, we implemented six control methods, where each returns either a solution derived from the ground truth data (positive control), a naive baseline prediction, or a randomly sampled prediction (negative control). The positive and negative control methods define an upper and lower bound for the performance metrics, which is used to normalize metric outputs. Full descriptions of the control methods can be found in **Appendix E.2.4**.

## 4 The Single-cell Perturbation Prediction Competition at NeurIPS 2023

To identify the state-of-the-art for perturbation prediction in unseen cell types, we hosted a Kaggle competition as part of the NeurIPS 2023 Competitions track called *Single-cell perturbation prediction: generalizing experimental interventions to unseen contexts*. This competition ran from September 12, 2023 through November 30, 2023 and used an earlier version of the dataset and benchmark before it was updated based on learnings from the competition (**Appendix B**). We ran the competition in two tracks. The Leaderboard Track followed the traditional data science competition setup with a public and private leaderboard tracking a single metric on public and private test sets (**Appendix B**). We also ran a Judges' Prize track where participants were judged based on a write-up addressing specific questions about perturbation prediction and the specific challenges of using our dataset to tackle this task. $50,000 in prizes were awarded for each track. The competition web page with the final leaderboard, code submissions, and discussions is available at: `https://www.kaggle.com/competitions/open-problems-single-cell-perturbations`

### 4.1 Leaderboard Track

In the leaderboard competition, competitors trained models on the training set and submitted CSV files with predictions for the public and private test tests. During the development phase (3 months), only the results from the public test set were used to calculate leaderboard rankings. During the final phase (5 days), competitors selected their top submission. Final scores were judged on the private test set only visible after the final submission deadline. Due to the limitations of the Kaggle platform, we ran the competition with a single metric, MRRMSE, decided on in collaboration with Kaggle data scientists. The participants were encouraged to use any publicly available external data.

Over the competition, 1,318 participants from 84 countries, forming 1,097 teams, submitted 25,529 solutions to our Leaderboard Track. This makes our competition one of the largest single-cell data science competitions to date. Although participants were only required to submit CSV predictions, the Kaggle platform has a strong culture of solution sharing. As such, we were able to read through reported submission code and identified trends among the best performing methods. We found that the top-scoring methods relied on diverse deep learning approaches, including transformer, LSTM, GRU, CNN and MLP architectures. The models used diverse loss functions, such as

mean squared error, mean absolute error, LogCosh ($L(y, \hat{y}) = \frac{1}{N} \sum_{i=1}^{N} \log(\cosh(y_i - \hat{y}_i))$), binary cross-entropy, MRRMSE, and Huber loss [29]. Despite several reported attempts, only the first of the three top-performing models relied on data other than the training set. The winning method used ChemBERTa [30], a pre-trained transformer, to encode the small molecule structure SMILES representation. According to the competitors' reports, data preprocessing proved to be very impactful. In particular, multiple competitors reported that target encoding and singular value decomposition of the high-dimensional input data were effective. One method used pseudolabels [31] for model training. All of the top three methods relied on model ensembles. We provide detailed descriptions of these methods in **Appendix E.1**.

## 4.2 Judges' Prize

In the Judges' Prize, participants were asked to address how biological priors or alternative model architectures influence leaderboard performance, to describe technical challenges that make perturbation prediction difficult, to characterize how data noise or downsampling affect model robustness, and to present well-documented and packaged model code. To identify winners, the write-ups were scored by a panel of single-cell experts. 17 teams submitted write-ups for a judges' prize, all of whom also participated in the leaderboard prize.

Many of the submissions provided valuable insights and were exceptionally detailed—the top-scoring team wrote a 33-page report. For example, several participants mentioned their efforts on integrating gene regulatory networks (GRN) inferred from ATAC and RNA data as an extra modality for prediction task [32, 33]. Although distinct patterns among cell types were observed from the provided ATAC-seq data, attempts at incorporating inferred GRNs in model predictions, even only for expression-enriched regulators, resulted in performance decreases in their models. Other groups attempted to use molecular interactions as an additional modality for model design. For example, GSEA-MsigDB [34] provides valuable information about pathways activated in various cell types. From these, a correlation network can be constructed based on shared pathways or shared regulation target genes. However, the models overall did not benefit from these efforts, which suggests that further filtering over inferred regulation/correlation relationships might be necessary. Finally, many submissions also investigated challenges associated with data representations and data pre-processing, which are described in the following section. We provide detailed descriptions of the Judges' Prize-winning methods in **Appendix E.2**.

## 4.3 Lessons learned

Here, we list several key learnings and opportunities to improve our benchmarking setup.

**False positives for unexpressed/lowly expressed genes**: DE analysis is sensitive to low-count genes, which can lead to overestimation of relative expression changes. This is especially problematic for compounds with subtle gene effects. To mitigate this, we employed a stricter gene filtering strategy per cell type [35], resulting in a reduced 5,317 genes (originally 18,211).

**Inconsistent annotations**: Proportions of T cell subtypes were inconsistent across donors (**Appendix Table 3**, **Appendix Figure 7**). These subtypes had low cell counts and subtle differences in expression that suggested misannotation, which may have been caused by perturbation impacts on marker gene expression. To resolve this, we grouped all of the T cells together in the final annotations **Figure 1b**.

**Outlier samples**: Certain samples had very few cells, which may be caused by perturbation-associated toxicity and was correlated with a high fraction of low $p$-values. To address this, we removed samples with $< 10$ cells or inconsistent cell type proportions across donors. We also removed three compounds for which we could not confidently annotate cell types (**Appendix F.2**), likely due to toxicity.

**Design matrix**: Due to a high number of factors and collinearity, the design matrix used in the competition (**Appendix Figure 6**) was not full-rank, potentially leading to parameter estimation issues. We updated the linear model to $f(g_j) = x_1 cc_i + x_2 p_i$, where $g_j$ is a gene, $cc_i$ is (cell type, compound) tuple, and $p_i$ is the plate. The resulting design matrix is full rank.

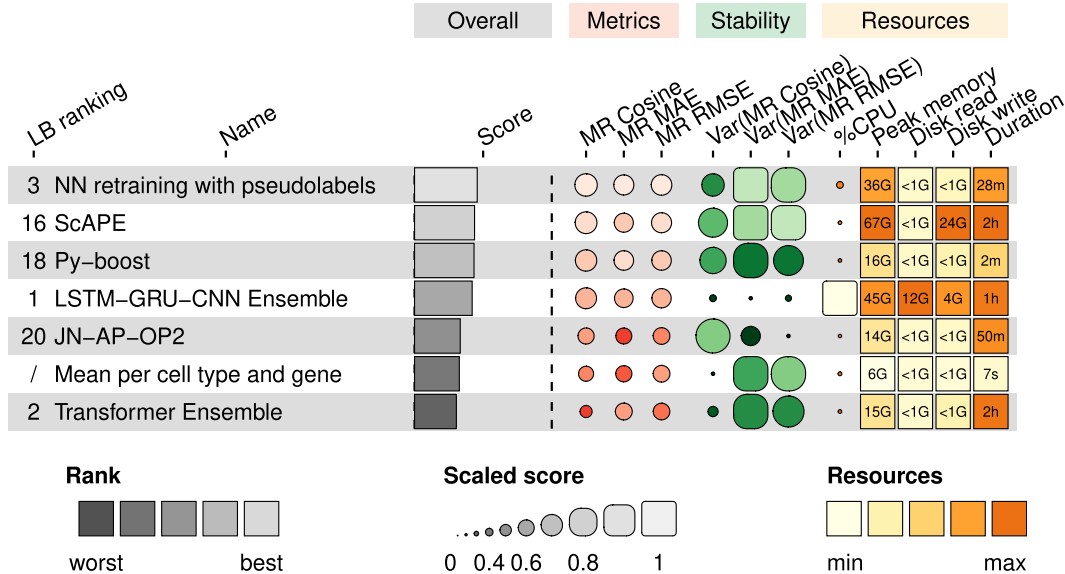

Figure 3: **An overview of the benchmarking results** of the six selected methods and one control method. Methods are ordered by the arithmetic mean of the three metrics. The MR Cosine, MR MAE, and MR RMSE were computed by comparing a method's predictions to the ground-truth data. Each of these metric values were min-max scaled between the positive control and random sample. The resources column group shows the resource usage of the various methods throughout their execution.

**Outlier $p$-values**: Our dataset contained some very low $p$-values (1e-180). As we do not want to penalize models for not differentiating between very small $p$-values, we clipped $p$-values in the dataset at 1e-4.

**Submit algorithms, not predictions**: Even though the competition participants submitted methods implementations, we were unable to exactly reproduce all of the results. We recommend requiring competitors to submit algorithms instead of predictions to promote the development of reusable tools. In addition, it allows algorithms to be more easily adapted, ultimately accelerating scientific discovery.

### 4.4 Updating the living benchmark

A central challenge in machine learning competitions is translating state-of-the-art methods according to competition leaderboards to impact applied science. A review of 10 years of machine learning competitions in dementia [36] found that no competition winners had been applied in clinical settings, suggested that winning methods may be overfitted to the competition dataset and metric, and suggested making methods available for testing in other settings. To enable further testing and evaluation of top-performing methods from our competition, we implemented and retrained the top 3 methods according to the leaderboard and the top 3 according to judges' scores in our Open Problems Perturbation Prediction living benchmark. This final benchmark includes the changes listed in the preceding section. Additionally, the public test set is now part of the training set. The results are shown in **Figure 3** and the latest results of the living benchmark are available at `https://openproblems.bio/results/perturbation_prediction`.

Examining model performance across compounds, we observed that for all 6 methods, the error residuals correlated with the number of DE genes. This indicates that the methods are better at predicting no change in gene expression than a significant change. Indeed, the top performing method, NN retraining with pseudolabels, predicts high $p$-values more often than they occur in the dataset (Appendix Figure **Figure 8**).

# 5   Discussion

In this study, we presented a living benchmark for single-cell perturbation prediction. The Open Problems Perturbation Prediction (OP3) benchmark features a newly generated fit-for-purpose dataset that is the largest of its kind, optimized data representations and metrics, positive and negative baseline methods that define performance ranges, and a cloud-based infrastructure that enables users to add new methods, metrics, and datasets to the benchmark. Using this benchmarking setup we ran the Single-cell Perturbation Prediction competition at NeurIPS 2023, in which over 1,300 participants contributed over 25,000 method solutions to address the challenge of predicting perturbation responses across drugs and cell types. This competition successfully made the topic of single-cell perturbation prediction accessible to a non-specialist community (more than half of the surveyed participants never worked with single-cell data **Appendix F.1**), while leveraging the expertise from this community to improve upon current state-of-the-art methods (via Leaderboard Track winners) and provide feedback on the task definition and implementation (via Judges' Prize winners). To promote the translation of competition outputs to domain impact in perturbation prediction, we used this competitor feedback to update the OP3 benchmark and populated it with the top-performing solutions from the competition. This enables methods to be further scrutinized by the community on the generalizability of their performance across data contexts and metrics.

To power our single-cell perturbation prediction competition and benchmark, we generated the largest multi-donor single-cell drug perturbation dataset on primary human tissue. However, despite profiling 146 drug perturbations in over 300,000 cells, the training data size is still limited from the perspective of building models that generalize across drugs, donors, and cell types. There are over 16,600 clinical-stage drugs [37], which typically elicit heterogeneous responses across cell types [38] and individuals [39]. Predicting the cell-type-specific response of a drug on an unseen individual will likely require data generation efforts that are not feasible by individual groups, but rather coordinated across consortia. Such efforts would also be needed to ensure aspects such as differing drug efficacy across genetic backgrounds [40, 41] can be taken into account, which is not feasible with existing perturbation datasets that often only profile cells from a single genotype [15]. In this context, our OP3 benchmark and dataset represent a first step towards this larger goal.

A further limitation of our competition, and indeed most other Kaggle competitions, derives from the use of a single performance metric, which is a limitation of the Kaggle platform. Goodhart's law suggests that when a performance metric becomes the optimization target, the metric ceases to be a good metric [42, 43]. This phenomenon is especially challenging when the chosen metric represents a proxy for good performance that is easy to evaluate during a model development loop (i.e. is differentiable and quickly calculable). In our case, perturbation prediction would ideally assess how well an unseen candidate drug treats a disease of unknown pathology in a particular patient. To make this tractable, we instead evaluate the transcriptome response in an unseen hold-out donor, cell type, and drug combination. A mitigation strategy for overfitting to this setting is to define additional relevant tasks related to perturbation prediction to evaluate method performance on different criteria. To promote innovation towards the overall goal of improving perturbation prediction, we specifically enable such a multi-task evaluation setup via the OP3 living benchmark and the design of our dataset. To promote generalizability of developed solutions [44], future competitions in this direction may further include orthogonal readouts, such as cell type proportions, rates of cell death, or inflammation [45].

Taken together, the OP3 benchmark and corresponding competition represent the first community-extensible standard for predicting perturbation responses from single-cell transcriptomic data. While several algorithms existed for this task also prior to our competition, the competition has been successful in greatly expanding the set of possible solutions available, which can be further scrutinized via the OP3 living benchmark. Indeed, the combination of a large-scale competition and a cloud-based living benchmark represents a promising approach to promoting innovation towards critical domain-specific challenges. We envision that the OP3 benchmark will lay the groundwork for further method development for this question, which is of critical importance to realize the promise of personalized medicine and optimized drug discovery.

## Acknowledgments and Disclosure of Funding

**Acknowledgements** We would like to thank Lijun Zhao, Roman Montez, Nicole Robichaud, Nina Colon, Sakina Saif, Laura Isacco, and Cameron Reilly at Cellarity for their contributions in generating the single-cell RNA sequencing data used in the publication. We also thank Yuge Ji for proofreading the manuscript. Saturn Cloud donated compute to support analysis of the winning methods.

**Funding** This work was supported by funds from the Chan Zuckerberg Initiative, Cellarity Inc., the Helmholtz Association and Helmholtz Munich. This work was co-funded by the European Union (ERC, DeepCell -101054957, to A.S. and F.J.T.). Views and opinions expressed are however those of the authors only and do not necessarily reflect those of the European Union or the European Research Council. Neither the European Union nor the granting authority can be held responsible for them.

**Competing interests** A.B., M.C., J.F., S.K., R.L., J.M-R., D.B. are paid employees of and have equity interest in Cellarity Inc. N.W. is a paid employee of and has equity interest in Olden Labs PBC. B.A. works for the US government. A.S. consults for Exvivo Labs Inc. R.M., J.T. are paid employees of and have equity interests in Retro Biosciences. F.J.T. consults for Immunai Inc., Singularity Bio B.V., CytoReason Ltd, Cellarity, and has ownership interest in Dermagnostix GmbH and Cellarity. M.D.L. contracted for the Chan Zuckerberg Initiative and received speaker fees from Pfizer and Janssen Pharmaceuticals.

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

# A Detailed dataset description

## A.1 Overview

We measure the impact of 146 compounds in human PBMCs with three replicates, one per each donor. The dataset was generated in a 96-well plate format with sample multiplexing such that each of the 12 wells in each row of the plate were pooled in a single lane of the 10x Chromium chip. We included controls in 3 out of 12 wells in each row of the plate such that each resulting multiplexed library contains a negative control DMSO well treatment and two positive controls of either belinostat or dabrafenib. The remaining 9 wells per row each contained a different treatment condition. The result is 576 unique scRNA samples (Appendix A.5). The dose of belinostat is $0.1\mu M$, DMSO $14.1\mu M$, and the rest of the compounds $1\mu M$. After sample demultiplexing, preprocessing, and quality control filtering, we retained 301,785 single cells and 21265 genes for further analysis. Further filtering and processing were performed to better tailor the dataset for the perturbation prediction task. Differential expression was computed with `limma` to create the representation of perturbation effects used in the benchmark (Appendix A.8).

## A.2 Data Availability

As is standard in the computational biology field, processed counts data is publicly available through the Gene Expression Omnibus (GEO) with accession GSE279945 and raw sequencing data is available through the Sequencing Read Archive (SRA) with accession PRJNA1149320.

**Maintenance plan**  The dataset will be stored on GEO and SRA indefenitely. Any updates to the dataset will be made available on these platforms. The souce code of the components and workflows used in this study are stored on GitHub at github.com/openproblems-bio/task_perturbation_prediction. At the time of publication, the project was published on GitHub and Zenodo as release `1.0.0`. Each component is backed by a Docker container published at ghcr.io/openproblems-bio, also using tag `1.0.0`. Any feedback or found errors can be reported through GitHub issues at `github.com/openproblems-bio/task_perturbation_prediction`.

**Responsibility**  We, the authors, bear all responsibility to withdraw our paper and data in case of violation of licensing or patient privacy rights. The dataset will be distributed under a Creative Commons license (CC BY 4.0).

## A.3 Culture of PBMCs

Human PBMCs from three donors were purchased from AllCells (`www.allcells.com`). Donor information is described in Table 1, and the informed donor consent process is described in Appendix J. PBMCs from one female and two males were used and were selected due to similarities in age and BMI, the absence of reported use of medications, and sufficient cell inventory for data generation.

Table 1: PBMC Donor Information

| Donor Name | Donor ID | Lot # | Age | Sex | BMI | Blood Type | Race | Smoker | CMV+ |
|---|---|---|---|---|---|---|---|---|---|
| Donor 1 | 110044355 | 3097601 | 45 | F | 25.4 | O+ | White | - | Neg |
| Donor 2 | 110044590 | 3096819 | 52 | M | 37.2 | O- | White | No | Pos |
| Donor 3 | 888676709 | 3094710 | 45 | M | 24.7 | A+ | White | No | Neg |

PBMCs were thawed in RPMI (Gibco cat # 11875-093) supplemented with 10% heat inactivated fetal bovine serum (HI-FBS, Gibco cat # 10082-147) and centrifuged for 8 minutes at 300 x G. The cell pellet was resuspended in RPMI supplemented with 10% HI-FBS, counted on a Luna fluorescent cytometer (Logos Biosystems) using AO/PI stain (Logos Biosystems, cat # F23001) per the manufacturer's instructions, and centrifuged to wash cells. The cell pellet was then resuspended to a concentration of 1,000,000 cells/mL in RPMI supplemented with 10% HI-FBS. For perturbation

experiments, cells were plated at 200,000 cells/well in 96-well V-bottom plates (Thermo Scientific cat # 277143) in 200 μL media and were cultured for a total of 48 hours prior to collection. For multiome profiling experiments, PBMCs were seeded into a T75 flask (Corning cat # 430641U) and were cultured 24 hours before collection.

## A.4   Characterization of PBMCs Across Donors

Flow cytometry was used to characterize the major cell populations in the PBMC samples from the three donors after thaw (0 hours) and at 48 hours of culture in 96-well V-bottom plates. This was performed to confirm the relative consistency of cell types across donors and to ensure that the time in culture and media conditions did not bias the survival of specific cell types. 200,000 PBMCs per well were seeded in a 96-well V-bottom plate and centrifuged for 8 minutes at 300 x G. The cell pellet was resuspended in antibodies against established cell lineage markers, which were diluted in Cell Staining Buffer (Biolegend cat # 420201) and incubated at 4C in the dark for 25 minutes. PBMCs incubated in a Cell Staining Buffer without adding antibodies were used as unstained controls. Cells were washed by centrifugation for 8 minutes at 300 x G and resuspended in a Cell Staining Buffer. Events were captured on a Novocyte Quanteon (config. V8B7Y6R4) with an average of 56,500 PBMCs per well acquired for analysis.

Prior to quantification, the spectral overlap of our conjugated antibodies was adjusted for using UltraComp eBeads™ Plus Compensation Beads (Invitrogen cat # 01-3333-42), per the manufacturer's instructions. Briefly, two conditions were used to compensate for spectral overlap: 1) unstained beads, and 2) single-colored controls with each antibody applied individually to the beads. Antibodies were incubated together with beads for 15 minutes, washed, and resuspended in a Cell Staining Buffer, following which events were captured on the cytometer. The compensation matrix was generated on FlowJo 10.8.1 and applied before the quantification of cell populations within PBMCs. The gating strategy used to quantify CD3+ T-cells, CD14+ myeloid cells, CD19+ B-cells, and CD56+ NK cells is described in Appendix Figure 4.

Overall, we observed that the four major cell populations measured were relatively consistent across all donors at each time point, with CD3+ T-cells comprising most cells within the sample. We noticed a reduction in the CD14+ myeloid compartment following the culture of the cells, which was consistent across all donors. We speculate this could be due to the myeloid population tending to differentiate and adhere following time in culture. We also acknowledge that the broad cell type markers used for characterization via flow cytometry do not permit quantification of more specific cell clusters (e.g., CD4+ vs CD8+ T-cells, monocytes vs. dendritic cells) that can be performed using gene markers in the sequencing data, making a direct comparison of cell numbers across the modalities more challenging. In sum, we selected PBMCs from three donors that contain relatively consistent numbers of cell types within each sample and perform similarly after 48 hours of culture.

## A.5   Compound information and treatment of PBMCs

146 compounds were applied to PBMCs from three healthy donors 24 hours after thawing and seeding into 96 well V-bottom plates. Compounds were selected based on two criteria:

1. Inclusion in Library of Integrated Network-Based Cellular Signatures (LINCS) Connectivity Map dataset, and
2. Robust transcriptional response observed in CD34+ hematopoietic stem cells (data not released).

These compounds also span a diverse range of mechanisms of action.

Compounds were resuspended in DMSO to 1 mM and arrayed onto a 96-well PCR plate. Each of the first three columns on the plate contained, respectively:

1. Belinostat, an HDAC inhibitor that we previously observed to induce a large transcriptional response in PBMCs (positive control).

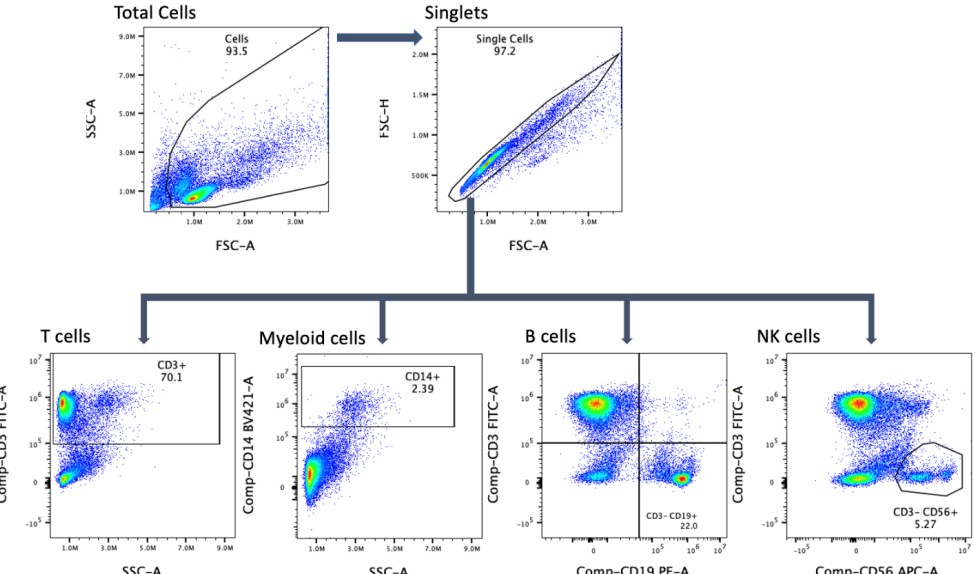

Figure 4: **Gating strategy for quantification of cell types within PBMCs.** FlowJo software was used to quantify the population of cells within PBMCs using the antibodies described in the methods section. First, a gate was generated by visualizing the forward scatter area and side scatter area to define the total live cells and to remove non-viable cells and debris from the analysis. Next, a gate was generated visualizing forward scatter height and area to define single cells and exclude doublets. Lastly, two-parameter density plots were used to assess the percentage of T cells (CD3+), myeloid cells (CD14+), B cells (CD3-, CD19+), and natural killer cells (CD3-, CD56+). Using this gating strategy, the percentage of cells within each population was quantified in PBMCs from the three donors used prior to full-scale data generation to ensure consistency in the samples and that time in cluster did not skew populations in a donor-specific manner.

2. Dabrafenib, a BRAF-inhibitor that we previously observed to induce a moderately-strong transcriptional response in PBMCs (positive control).

3. No compound treatment except for DMSO (negative control).

Each well in columns 4-12 of the 96-well PCR plate (72 wells) contained a unique treatment compound. On the day of treatment, compounds were diluted and mixed directly in the PCR to 100 μM in RPMI (Gibco cat # 11875-093) using an Integra Viaflo 384 automated liquid handler. 2 μL of compound in solution was then transferred using the Integra Viaflo 384 automated liquid handler from the PCR plate and applied to PBMCs cultured in a 96-well V-bottom plate, described above. The use of an automated liquid handler enabled simultaneous application of 96 compounds and limited errors in transferring. The final concentration of compound applied to the cells was 1 μM. Cells were treated 24 hours before collection.

**A.6   Single-cell sequencing of perturbed PBMCs**

48 hours after thaw, PBMCs cultured and treated in 96-well V-bottom plates were mixed with an Integra Viaflo 384, and a sample of cell suspension was transferred into a Thermo Scientific U-bottom plate (cat # 163320) containing CountBright Plus Absolute Counting Beads (Invitrogen cat # C36995) and SYTOX AADvanced Ready Flow (Invitrogen cat # R37173) diluted in DPBS, per the manufacturer's instructions. Total viability and live cell number per well were quantified via flow cytometry using a BD FACSCelesta Cell Analyzer (BD Biosciences).

The remaining treated PBMCs were centrifuged in the 96 well V-bottom plate at 300 x G for 8 minutes. Culture media was aspirated using semi-automated liquid handling to not disturb the cell pellet and washed once with Cell Staining Buffer. Cells were centrifuged and resuspended in 12

unique Cell Multiplexing Oligonucleotides (3' CellPlex Kit Set A, 10X Genomics, cat # 1000261) applied in columns 1 through 12, incubated for 5 minutes at room temperature, and then quenched and washed three times in DPBS supplemented with 4% human serum albumin (Grifols cat # NDC 68516-5216-1). This allowed for consolidation of a 96-well plate into 8 pools, each containing cells from a well labeled with a unique Cell Multiplexing Oligonucleotide. To ensure equal sequencing representation from each compound treatment, 100,000 cells per well (calculated from the initial cell count) were pooled by row into a 5 mL conical tube, resulting in a total of 8 pools, 1 conical tube collected per row, using an Integra Assist Plus and associated D-ONE module automated liquid handling instrument (Integra Biosciences).

Final pools therefore contained a DMSO negative control, 2 transcriptionally active positive controls, and 9 experimental compounds. Cells were pelleted by centrifugation and resuspended to 1.2 x 106 cells/mL in Cell Staining Buffer for downstream preparation for single-cell sequencing. Single-cell libraries were prepared from each pool using Chromium Next GEM Single-cell 3' Kit v3.1 (10X Genomics, cat # 1000268) following the manufacturer's recommended protocol (10X Genomics, CG000388 Rev C).

Briefly, a total of 12,000 cells (1,200 per multiplexing oligo) were loaded into a single channel of a Chromium Next GEM Chip G (10X Genomics, Cat # 1000120) and partitioned into droplets with gel beads using a Chromium controller (10X Genomics, cat # 1000204). After emulsion droplets were formed and collected, reverse transcription reactions were incubated at 53C for 45 minutes. Barcoded transcripts were purified, amplified and size fractionated to create separate libraries for the transcriptome and feature barcode fractions from each sample. Transcriptome libraries were fragmented and ligated to indexed sequencing adapters according to the manufacturer's recommended protocol. Feature barcode libraries were prepared using 3' Feature Barcode Kit (10X Genomics, cat # 1000262) following the manufacturer's recommended protocol (10X Genomics, CG000388 Rev C). Transcriptome libraries were sequenced on an Illumina NovaSeq6000 with paired end reads as follows: Read 1 = 28 cycles, i7 Index = 10 cycles, i5 = 10 cycles, Read 2 = 89 cycles. Feature barcode libraries were sequenced on an Illumina NovaSeq6000 with paired end reads as follows: Read 1 = 28 cycles, i7 Index = 10 cycles, i5 = 10 cycles, Read 2 = 35 cycles. Cell Ranger (v5.0.1) mkfastq was used to generate all demultiplexed FASTQ files from the raw sequencing data.

Cell Ranger count was used to align transcriptome reads to the human GRCh38 genome reference, identify corresponding feature barcode reads according to a csv reference file containing all the relevant information needed for downstream processing, and quantify gene and UMI counts.

### A.7 Multiome ATAC + gene expression profiling of PBMCs at baseline

24 hours after thaw, PBMCs cultured in T75 flasks were collected in a 50 mL conical tube, centrifuged at 300 x G for 8 minutes, and washed once with DPBS. Viability and total live cells/mL were quantified using AO/PI stain on a Luna fluorescent cytometer. Nuclei were isolated from cells using Chromium Next GEM Single-cell Multiome ATAC + Gene Expression Reagent Bundle (10X Genomics, cat # 1000283) and Chromium Nuclei Isolation Kit with RNase Inhibitor (10X Genomics, cat # 1000494) following the manufacturer's recommended protocol (10X Genomics, CG000365, Rev C). Briefly, 1.2 million cells were pelleted and resuspended in 100 μL of lysis buffer and incubated on ice for 5 minutes. Multiple rounds of washing were followed by resuspension in 150 μL of diluted nuclei buffer and filtered through a 40 μm Flowmi Cell Strainer (Fisher Scientific, cat # 14100150). Nuclei were counted on a Nexcelom Cellometer K2.

Mutliome ATAC + Gene Expression libraries were prepared using Chromium Next GEM Single-cell Multiome ATAC + Gene Expression Reagent Bundle (10X Genomics, cat # 1000283) following the manufacturer's recommended protocol (10X Genomics, CG000338, Rev F). Briefly, a total of 8,000 nuclei were targeted for loading into a transposition reaction, which incubated at 37 C for 60 minutes. The output was then loaded into a single channel of a Chromium Next GEM Chip J (10X Genomics, cat # 1000234) and partitioned into droplets with gel beads using a Chromium controller (10X Genomics, cat # 1000204). After emulsion droplets were formed and collected,

reverse transcription and transposed DNA fragment barcoding reactions were incubated at 37C for 45 minutes. Both products were purified, amplified and size fractionated to create the ATAC and transcriptome fractions from each sample. ATAC libraries had indexed sequencing adapters added. Transcriptome libraries were fragmented and ligated to indexed sequencing adapters. ATAC libraries were sequenced on an Illumina NovaSeq6000 with paired end reads as follows: Read 1 = 50 cycles, i7 Index = 8 cycles, i5 = 24 cycles, Read 2 = 49 cycles. Transcriptome libraries were sequenced on an Illumina NovaSeq6000 with paired end reads as follows: Read 1 = 28 cycles, i7 Index = 10 cycles, i5 = 10 cycles, Read 2 = 89 cycles. Cell Ranger ARC (v2.0) mkfastq was used to generate all demultiplexed FASTQ files from the raw sequencing data. Cell Ranger ARC count was used to align transcriptome reads to the human GRCh38 genome reference provided by 10X Genomics and generate all downstream count matrices.

## A.8 Processing of scRNA-seq perturbation data

Starting with the counts matrix output from Cell Ranger, cells with total numbers of transcripts that fell below or above specific thresholds were filtered out of the dataset. These transcript thresholds were determined per sequencing pool. All cells that had a mitochondrial counts fraction higher than 0.2 were also removed. The Python package `scrublet` was then used to label cells with a probability of being doublets. These probabilities were smoothed over a $k$-nearest neighbor graph constructed from the cells, and cells with a smoothed doublet probability of greater than 0.2 were filtered out.

During the pooling process (Appendix A.6), cells from each of the twelve wells in a plate row were tagged with distinct cell multiplexing oligonucleotides to increase throughput and decrease batch effects across wells. This multiplexing procedure necessitated a demultiplexing step in the processing pipeline, whereby a multivariate Gaussian-mixture model was used to identify the well from which each cell most likely originated. Cells that could not be conclusively labeled as belonging to any particular well were dropped from the dataset.

Prior to cell-type annotation, counts were normalized to sum to 1000 in each cell and then transformed with the mapping $x \mapsto \ln(x + 1)$. Cell-type annotation was performed by first running Leiden clustering with `resolution = 1` on a $k$-nearest neighbor graph built from the 2000 most highly-variable genes, then manually assigning a cell type label (T-cells, B-cells, myeloid cells, or NK cells) to each cluster based on expression of the marker genes in Table 2. In addition, we filtered samples of certain compounds as described in Appendix F.2.

Table 2: PBMC Marker Genes

| Cell Type | Marker Genes |
|---|---|
| T-cells | CCR6, CCR7, CD2, CD27, CD3D, CD3E, CD3G, CD4, CD6, CD8A, CTLA4, FOXP3, GZMB, IL2RA, PTPRC, TRDV2, TRGC1, TRGV9 |
| B-cells | CD19, CD24, CD24, CD27, CD38, CD38, CD38, CD79A, CD79B, DERL3, FKBP11, HLA-DQA1, HLA-DQB1, IGLL5, IGLL5, IGLL5, JCHAIN, MS4A1, PAX5, PRDX4, PTPRC, SEC11C, SSR4, TCL1A, VPREB3 |
| Myeloid cells | CD14, CD163, CD1C, CD68, CD83, ITGAX |
| NK cells | CD2, CD69, COX6A2, FCGR3A, GNLY, GZMA, GZMB, GZMM, KIR2DL4, KLRB1, NCAM1, NCR1, NKG7, NKG7, ZMAT4 |

Next, differential expression (DE) was performed to produce a representation of the perturbation effects of each drug. To ensure our DE computation was as robust as possible, we used the `filterByExpr` function from the `EdgeR` package to filter down to 5317 genes that were consistently expressed across all cell types. Counts from these 5317 genes are then summed across the cells of each type in every well to produce what is known as a *pseudobulked* expression object. The

pseudobulked counts are then fed into the `limma`/`voom` pipeline to compute moderated $p$-values and log-fold change statistics per gene for each condition. The linear model fit by `limma` included an additional covariate that captured the plate-to-plate batch effects. This covariate also reflected the variability in perturbation effect across donors, as each plate contained samples from only one donor.

All of the processing steps described in this section, unless explicitly stated otherwise, were performed using the `scanpy` library [46].

## A.9 Processing of baseline Multiome snRNA-seq/scATAC-seq data

For processing the joint snRNA-seq/scATAC-seq measurements, we start with the outputs provided by the Cell Ranger pipeline, namely:

1. the fragments file, which lists both the region of the chromosome and the cell barcode for each detected ATAC-seq fragment, and

2. the filtered feature-barcode matrix, which contains both the detected genes (snRNA-seq) and called peaks (scATAC-seq) assigned to each cell barcode.

We first split up the feature-barcode matrix into a cell-by-gene snRNA-seq matrix and a cell-by-peaks scATAC-seq matrix. The QC steps for the snRNA-seq measurements were nearly identical to the process described in Appendix A.8 for the scRNA-seq data, albeit with a slight hand-tuning of the filtering thresholds. Namely, cells with low counts (below 500 transcripts), high mitochondrial counts percentage (above 0.2), or high probability of being doublets (above 0.2) were removed, and genes that were expressed in fewer than 100 cells were also filtered out, resulting in 17438 distinct genes. Following this, counts were normalized to sum to 1000 in each cell and then rescaled using the mapping $x \mapsto \ln(x + 1)$.

Cells were further filtered using the scATAC-seq measurements. Specifically, cells that met any of the following criteria were removed:

1. fewer than 1000 fragments,

2. fewer than 750 peaks,

3. *transcription start site (TSS) enrichment score* below 0.8, or

4. *nucleosome signal* below 2.0.

Both the TSS enrichment score and nucelosome signal are common metrics for evaluating the quality of chromatin accessibility measurements. The TSS enrichment score is calculated by taking windows of 2000bp around either side of TSSs, then computing the average ratio of read depth at 100bps on either side of these windows to the read depth at the respective TSS in the center of the window [47]. For the sake of computational efficiency, we estimate the TSS enrichment score by computing this average ratio over a random subset of 3000 TSSs rather than every TSS.

The nucleosome signal is the ratio of the number of single-nucleosome fragments (between 147bp and 294bp) to the number of nulceosome-free fragments (shorter than 147bp). Again for the sake of computational efficiency, we estimate the the nucleosome signal using a subset of the ATAC-seq fragments.

Specific peaks that were observed in fewer than 20 cells were also dropped.

After filtering, cell type annotation was performed per-donor by running Leiden clustering, then assigning all the cells in each cluster a cell type label using `celltypist` [48]. These annotations were then validated based on the expression of the marker genes listed in Table 2. If a cluster could not be conclusively labeled with a specific cell type, the cells from that cluster were filtered out. All of these preprocessing steps were performed with either `scanpy` (for snRNA-seq) and `muon` (for scATAC-seq) [46, 49].

### A.10 Datasheet for datasets

| Motivation |
|---|

**For what purpose was the dataset created?** Was there a specific task in mind? Was there a specific gap that needed to be filled? Please provide a description.

OP3 dataset was created to enable research on predicting cell-type specific transcriptomic response to drugs. The dataset was created intentionally with that task in mind, providing donor replicates to account for the variability of outcomes.

**Who created this dataset (e.g., which team, research group) and on behalf of which entity (e.g., company, institution, organization)?**

The authors of this paper, along with the additional scientists at Cellarity listed in the acknowledgment section, namely Lijun Zhao, Roman Montez, Nicole Robichaud, Nina Colon, Sakina Saif, Laura Isacco, and Cameron Reilly.

**Who funded the creation of the dataset?** If there is an associated grant, please provide the name of the grantor and the grant name and number.

This work was supported by funds from the Chan Zuckerberg Initiative, Cellarity Inc., the Helmholtz Association and Helmholtz Munich. This work was co-funded by the European Union (ERC, DeepCell -101054957).

**Any other comments?**

None.

| Composition |
|---|

**What do the instances that comprise the dataset represent (e.g., documents, photos, people, countries)?** Are there multiple types of instances (e.g., movies, users, and ratings; people and interactions between them; nodes and edges)? Please provide a description.

OP3 contains scRNA-seq data of PBMCs across three donors. Cells are either control (DMSO) or were exposed to one of 146 drugs. It also provides multimodal (joint snRNA-seq and scATAC-seq) data for the same three donors at baseline. Processed data contains $p$-values and log-fold change per cell type and gene for each drug, which indicate the significance and magnitude of gene expression change induced by a given compound in a given cell type.

**How many instances are there in total (of each type, if appropriate)?**

There are 449650 cells collected across 576 samples in the raw scRNA-seq dataset. After filtering for the perturbation prediction task, this becomes 298087 cells across 567 samples.

Meanwhile, the raw multiome snRNA-seq/scATAC-seq data contains 53086 cells, which are filtered down to 22591 during processing.

**Does the dataset contain all possible instances or is it a sample (not necessarily random) of instances from a larger set?** If the dataset is a sample, then what is the larger set? Is the sample representative of the larger set (e.g., geographic coverage)? If so, please describe how this representativeness was validated/verified. If it is not representative of the larger set, please describe why not (e.g., to cover a more diverse range of instances, because instances were withheld or unavailable).

While individual cells and samples were removed from the raw data for failing to pass quality-control, the raw data is available to download and represents all the samples that were collected in this experiment.

**What data does each instance consist of? "Raw" data (e.g., unprocessed text or images) or features?** In either case, please provide a description.

The most raw form of the data is a collection of `.bcl` files from the Illumina sequencer (not released). These are then processed into `.fastq` files, which we have made available on the Sequencing Read Archive (SRA), as is standard practice for the computational biology field. `.fastq` files are then converted into raw counts matrices through standard Cell Ranger bioinformatics pipelines. For

the scRNA-seq data, the combined raw counts matrix has 449650 rows (cells) and 58676 columns (genes). The majority of columns contain either all zeros or very few measuremnts. For multimodal snRNA-seq/scRNA-seq data, the raw counts matrix has 53086 rows (cells) and 172019 columns. Of these columns, 36601 are gene expression measurements, while the other 135418 measure chromatin accessibility. Similar to the scRNA-seq data, this matrix is extremely sparse.

**Is there a label or target associated with each instance?** If so, please provide a description.

The only information that is known about any given cell with absolute certainty is which sequencing library, plate, and donor the cell originated from. However, if a cell can be assigned to a given well in the demultiplexing process (Appendix A.8), then well-level metadata, which includes compound treatment, can be attached to the cell. Moreover, marker gene expression can be used to label the majority of cells with high-confidence cell type annotations.

The processed dataset (after DGE analysis) contains a $-\log_{10}(p\text{-value}) \times \text{sign}(\text{log-fold change})$ statistic for each cell type, compound, and gene, which indicates the significance and direction of a gene expression change.

**Is any information missing from individual instances?** If so, please provide a description, explaining why this information is missing (e.g., because it was unavailable). This does not include intentionally removed information, but might include, e.g., redacted text.

Single-cell RNA-seq data is sparse, meaning that counts for the majority of genes are missing from each individual cell. This sparsity is caused by a number of different factors, including stochasticity of gene expression and constraints on read depth per cell. In addition, the wells with certain compounds had few or no viable cells to sequence, which might have been a result of compound toxicity or experimental conditions in a given well.

**Are relationships between individual instances made explicit (e.g., users' movie ratings, social network links)?** If so, please describe how these relationships are made explicit.

Which cells belong to the same donor or were cultured on the same plate can be determined directly from the raw data. Among the cells that can be successfully demultiplexed (Appendix A.8), it can be further determined which cells came from the same well and which were treated with the same compound.

**Are there recommended data splits (e.g., training, development/validation, testing)?** If so, please provide a description of these splits, explaining the rationale behind them.

See Appendix B.

**Are there any errors, sources of noise, or redundancies in the dataset?** If so, please provide a description.

See Appendix I.2.

**Is the dataset self-contained, or does it link to or otherwise rely on external resources (e.g., websites, tweets, other datasets)?** If it links to or relies on external resources, a) are there guarantees that they will exist, and remain constant, over time; b) are there official archival versions of the complete dataset (i.e., including the external resources as they existed at the time the dataset was created); c) are there any restrictions (e.g., licenses, fees) associated with any of the external resources that might apply to a future user? Please provide descriptions of all external resources and any restrictions associated with them, as well as links or other access points, as appropriate.

The dataset is entirely self-contained.

**Does the dataset contain data that might be considered confidential (e.g., data that is protected by legal privilege or by doctor-patient confidentiality, data that includes the content of individuals non-public communications)?** If so, please provide a description.

The dataset contains human samples that were obtained with the consent of the subjects. See Appendix J.

**Does the dataset contain data that, if viewed directly, might be offensive, insulting, threatening, or might otherwise cause anxiety?** If so, please describe why.

No.

**Does the dataset relate to people?** If not, you may skip the remaining questions in this section.

Yes. The data was derived from human blood samples.

**Does the dataset identify any subpopulations (e.g., by age, gender)?** If so, please describe how these subpopulations are identified and provide a description of their respective distributions within the dataset.

Yes, we included the age, sex, BMI, race, smoker status, and CMV+ status of the donors. The data was collected through the general health interview described in Appendix J.

**Is it possible to identify individuals (i.e., one or more natural persons), either directly or indirectly (i.e., in combination with other data) from the dataset?** If so, please describe how.

See Appendix J.

**Does the dataset contain data that might be considered sensitive in any way (e.g., data that reveals racial or ethnic origins, sexual orientations, religious beliefs, political opinions or union memberships, or locations; financial or health data; biometric or genetic data; forms of government identification, such as social security numbers; criminal history)?** If so, please provide a description.

The data contains the racial origin and health information, including BMI, smoker status, and CMV+ status of the donors that were collected through the general health interview described in Appendix J. In theory, unique gene expression patterns could be used to identify donors.

**Any other comments?**

None.

| Collection Process |
| --- |

**How was the data associated with each instance acquired?** Was the data directly observable (e.g., raw text, movie ratings), reported by subjects (e.g., survey responses), or indirectly inferred/derived from other data (e.g., part-of-speech tags, model-based guesses for age or language)? If data was reported by subjects or indirectly inferred/derived from other data, was the data validated/verified? If so, please describe how.

We performed the scRNA-seq and multimodal snRNA-seq/scATAC-seq assays to study the effects of the drugs on the gene expression, as described in Appendix A.1.

**What mechanisms or procedures were used to collect the data (e.g., hardware apparatus or sensor, manual human curation, software program, software API)?** How were these mechanisms or procedures validated?

The experiments are described in detail in Appendix A.

**If the dataset is a sample from a larger set, what was the sampling strategy (e.g., deterministic, probabilistic with specific sampling probabilities)?**

The raw data is available to download and represents all the samples that were collected in this experiment.

**Who was involved in the data collection process (e.g., students, crowdworkers, contractors) and how were they compensated (e.g., how much were crowdworkers paid)?**

The information on sample collection is available in Appendix J.

**Over what timeframe was the data collected? Does this timeframe match the creation timeframe of the data associated with the instances (e.g., recent crawl of old news articles)?** If not, please describe the timeframe in which the data associated with the instances was created.

Cells were collected from patients in 2021-2022, while the perturbation experiments were performed at Cellarity in June and July of 2023.

**Were any ethical review processes conducted (e.g., by an institutional review board)?** If so, please provide a description of these review processes, including the outcomes, as well as a link or other access point to any supporting documentation.

See Appendix J.

**Does the dataset relate to people?** If not, you may skip the remaining questions in this section.

Yes.

**Did you collect the data from the individuals in question directly, or obtain it via third parties or other sources (e.g., websites)?**

We purchased commercially available human tissue samples from AllCells, Inc.

**Were the individuals in question notified about the data collection?** If so, please describe (or show with screenshots or other information) how notice was provided, and provide a link or other access point to, or otherwise reproduce, the exact language of the notification itself.

Yes, see Appendix J.

**Did the individuals in question consent to the collection and use of their data?** If so, please describe (or show with screenshots or other information) how consent was requested and provided, and provide a link or other access point to, or otherwise reproduce, the exact language to which the individuals consented.

Yes, see Appendix J.

**If consent was obtained, were the consenting individuals provided with a mechanism to revoke their consent in the future or for certain uses?** If so, please provide a description, as well as a link or other access point to the mechanism (if appropriate).

Yes, see Appendix J.

**Has an analysis of the potential impact of the dataset and its use on data subjects (e.g., a data protection impact analysis) been conducted?** If so, please provide a description of this analysis, including the outcomes, as well as a link or other access point to any supporting documentation.

See Appendix J.

**Any other comments?**

None.

| Preprocessing/cleaning/labeling |
|---|

**Was any preprocessing/cleaning/labeling of the data done (e.g., discretization or bucketing, tokenization, part-of-speech tagging, SIFT feature extraction, removal of instances, processing of missing values)?** If so, please provide a description. If not, you may skip the remainder of the questions in this section.

We provide the raw version of the dataset, processed, and the code used for data processing. Data processing is described in Appendix A.8 and A.9.

**Was the "raw" data saved in addition to the preprocessed/cleaned/labeled data (e.g., to support unanticipated future uses)?** If so, please provide a link or other access point to the "raw" data.

Raw data for both the perturbational scRNA-seq and baseline snRNA-seq/scATAC-seq data are currently available through the Sequencing Read Archive (SRA) with accession PRJNA1149320.

**Is the software used to preprocess/clean/label the instances available?** If so, please provide a link or other access point.

Yes, see `github.com/theislab/task-dge-perturbation-prediction-analysis` and `github.com/openproblems-bio/task_perturbation_prediction` for the code used for data processing. In addition, other steps not included in the code are outlined in Appendix A.

**Any other comments?**

None.

| Uses |
|---|

**Has the dataset been used for any tasks already?** If so, please provide a description.

The dataset has been used for the Kaggle competition as part of the NeurIPS 2023 Competitions track called *Single-cell perturbation prediction: generalizing experimental interventions to unseen contexts*. It was also used to develop the benchmark described in this paper, see Section 3.

**Is there a repository that links to any or all papers or systems that use the dataset?** If so, please provide a link or other access point.

The dataset will be officially released with this publication. Hence, no other papers used this dataset.

**What (other) tasks could the dataset be used for?**

Aside from the use outlined in this study, the dataset enables myriad other inquiries, including but not limited to: the impact of different compound mechanisms of action on gene expression, the variance in compound effects across donors, pathway-based analyses of perturbation effects, etc.

**Is there anything about the composition of the dataset or the way it was collected and preprocessed/cleaned/labeled that might impact future uses?** For example, is there anything that a future user might need to know to avoid uses that could result in unfair treatment of individuals or groups (e.g., stereotyping, quality of service issues) or other undesirable harms (e.g., financial harms, legal risks) If so, please provide a description. Is there anything a future user could do to mitigate these undesirable harms?

Some medical information in the dataset could be used for deanonymization. However, given the limited scope of the provided data, it is highly unlikely that particular individuals or groups would be unfairly treated as a result of using this dataset.

**Are there tasks for which the dataset should not be used?** If so, please provide a description.

Given the limited scope of this dataset, it should not be used to influence immediate medical decision-making.

**Any other comments?**

None.

---

| Distribution |
|:---:|

**Will the dataset be distributed to third parties outside of the entity (e.g., company, institution, organization) on behalf of which the dataset was created?** If so, please provide a description.

Yes, the dataset will be publicly available on the internet.

**How will the dataset will be distributed (e.g., tarball on website, API, GitHub)** Does the dataset have a digital object identifier (DOI)?

As is standard in the computational biology field, processed counts data is publicly available through the Gene Expression Omnibus (GEO) with accession GSE279945 and raw sequencing data is available through the Sequencing Read Archive (SRA) with accession PRJNA1149320.

**When will the dataset be distributed?**

If this paper is accepted into the Datasets and Benchmarks track, the dataset will be distributed publicly with the submission of the camera-ready version of the paper, at the latest. However, we will likely release the dataset sooner due to interest in the single-cell research community.

**Will the dataset be distributed under a copyright or other intellectual property (IP) license, and/or under applicable terms of use (ToU)?** If so, please describe this license and/or ToU, and provide a link or other access point to, or otherwise reproduce, any relevant licensing terms or ToU, as well as any fees associated with these restrictions.

The dataset will be distributed under a Creative Commons license (CC BY 4.0).

**Have any third parties imposed IP-based or other restrictions on the data associated with the instances?** If so, please describe these restrictions, and provide a link or other access point to, or otherwise reproduce, any relevant licensing terms, as well as any fees associated with these restrictions.

No.

**Do any export controls or other regulatory restrictions apply to the dataset or to individual instances?** If so, please describe these restrictions, and provide a link or other access point to, or otherwise reproduce, any supporting documentation.

No.

**Any other comments?**

None.

| Maintenance |
|:---:|

**Who will be supporting/hosting/maintaining the dataset?**

The authors of this paper. The dataset will be hosted on the GEO platform indefinitely.

**How can the owner/curator/manager of the dataset be contacted (e.g., email address)?**

Contact the last author of this paper (dburkhardt@cellarity.com).

**Is there an erratum?** If so, please provide a link or other access point.

No.

**Will the dataset be updated (e.g., to correct labeling errors, add new instances, delete instances)?** If so, please describe how often, by whom, and how updates will be communicated to users (e.g., mailing list, GitHub)?

If any correction is needed, such as adjustments to metadata or refiltering of cells, we will upload a new version of the dataset to GEO. This will be noted on the OP3 benchmark GitHub page.

**If the dataset relates to people, are there applicable limits on the retention of the data associated with the instances (e.g., were individuals in question told that their data would be retained for a fixed period of time and then deleted)?** If so, please describe these limits and explain how they will be enforced.

There is no such limit. See Appendix J.

**Will older versions of the dataset continue to be supported/hosted/maintained?** If so, please describe how. If not, please describe how its obsolescence will be communicated to users.

Older versions will be available to download on GEO.

**If others want to extend/augment/build on/contribute to the dataset, is there a mechanism for them to do so?** If so, please provide a description. Will these contributions be validated/verified? If so, please describe how. If not, why not? Is there a process for communicating/distributing these contributions to other users? If so, please provide a description.

Changes to data postprocessing can be proposed in GitHub issues and Pull Requests at github.com/openproblems-bio/task_perturbation_prediction. For all other changes, contact the authors of the paper.

**Any other comments?**

None.

## B   Data splits

To derive the competition training and test splits, the compounds were divided into three groups, public train, public test, and private test, at a ratio of 1:3:5. This lopsided train-test split was chosen to determine whether we could model perturbation signatures in unseen cell types while only measuring roughly 10% of the compounds in those cell types. Differential expression values were provided to competitors for all cell types for compounds in the train set but masked in B and myeloid cells for test perturbations, although they could evaluate their models against the public test set an unlimited number of times. The final score was computed only on the private test set.

To avoid data leakage from the test set, we fit the training and test set DE models separately. To generate the training data, we fit the DE model on only the samples from the training set. To generate the private and public test data, we fit the DE model to all samples in the experiment. This kept the test data private and ensured the test data was the most accurate.

For the benchmark, we use only two splits, train and test, where the train split contains public train and public test data, and the test split contains only private test data.

## C  Benchmarking representations of perturbation effect

### C.1  Cross-donor retrieval

As mentioned in Section 3.3, we developed the cross-donor retrieval heuristic to compare different representations of perturbation effects. This heuristic is calculated as follows. Let:

1. $C = \{c_1, \ldots, c_{140}\}$ be the list of compounds,

2. $G = \{g_1, \ldots, g_{5317}\}$ be the list of genes,

3. $T = \{t_1, t_2, t_3, t_4\}$ be the list of cell types, and

4. $D = \{d_1, d_2, d_3\}$ be the list of donors.

First, we compute differential expression (DE) across all genes for each donor-compound-cell type combination $(d, c, t)$. Note that this is slightly different from the approach we take in computing DE for the task data. In that context, we include data from all donors in our model and then add a donor covariate to regress out donor-specific effects. For computing cross-donor retrieval, we compute DE for each donor separately[1].

We let $\mathrm{pert}_{d,c,t,g} \in \mathbb{R}$ denote the representation for gene $g$ of the perturbation $(d, c, t)$, and let

$$\mathrm{pert}_{d,c,t} \in \mathbb{R}^{|G|}$$

be the vector of representations for all genes. The for a fixed donor pair $(d_i, d_j)$ and cell type $t_k$ we compute the pairwise distance matrix

$$\begin{bmatrix} \left\| \mathrm{pert}_{d_i,c_1,t_k} - \mathrm{pert}_{d_j,c_1,t_k} \right\| & \cdots & \left\| \mathrm{pert}_{d_i,c_1,t_k} - \mathrm{pert}_{d_j,c_{140},t_k} \right\| \\ \vdots & \ddots & \vdots \\ \left\| \mathrm{pert}_{d_i,c_{140},t_k} - \mathrm{pert}_{d_j,c_1,t_k} \right\| & \cdots & \left\| \mathrm{pert}_{d_i,c_{140},t_k} - \mathrm{pert}_{d_j,c_{140},t_k} \right\| \end{bmatrix}.$$

Now we replace each value in this pairwise distance matrix with its (ascending) rank among the values in the same row. After computing the ranked distance matrix for all three pairs of donors, we extract the diagonals of these matrices. This distribution of values for various representations and metrics can be seen in Figure 2 of the main paper.

### C.2  Perturbation effect representation

In Figure 2, we compared the following representations:

1. $\log(\mathrm{counts} + 1)$: natural log of raw counts per gene, with an additional pseudocount to prevent taking the logarithm of 0.

2. log-fold change: base-2 logarithm of change in normalized gene expression under the effect of a perturbation, taken directly from the `logFC` output from `limma`.

3. $p$-value: significance of change of gene expression, taken directly from the `P.Value` output from `limma`.

---

[1]While there is only one well for all treatment compounds per donor (besides the positive controls), there are 16 negative control wells for each donor. Hence, we can obtain estimates for the statistical significance of perturbation effects by comparing gene expression in the 1 treatment well against the 16 negative control wells.

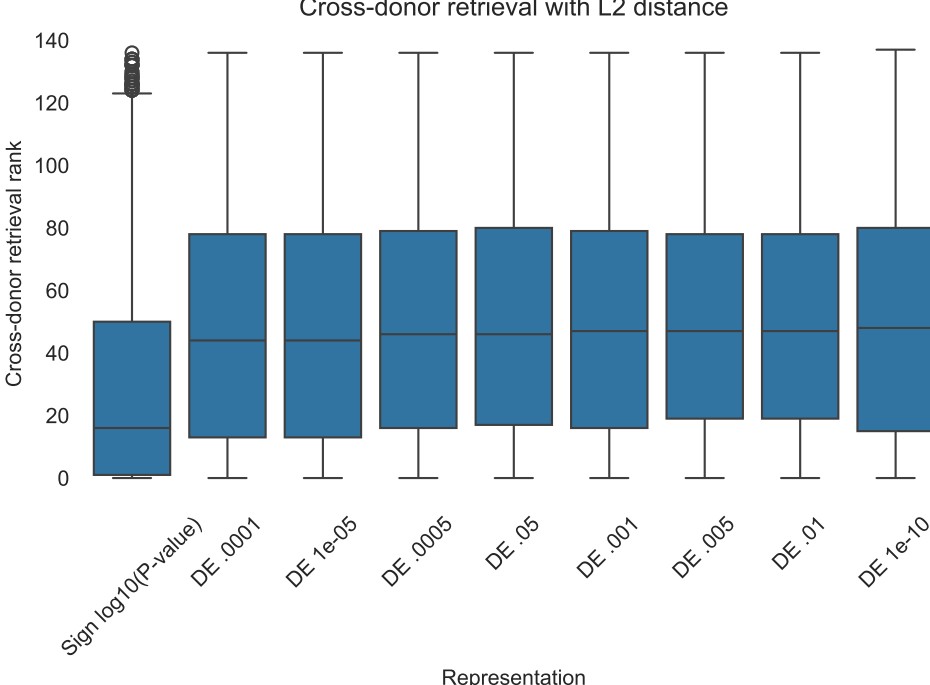

Figure 5: **Cross-donor retrieval for binarized significance.** "DE X" stands for the binarized representation, with sign indicating direction, and X indicates the threshold of significance.

4. $-\log_{10}(p\text{-value}) \times \text{sign}(\text{log-fold change})$: the magnitude of this value correlates with the statistical significance of the change in gene expression, while the sign corresponds to the direction of the change.

5. $-\log_{10}(\text{FDR-adjusted } p\text{-value}) \times \text{sign}(\text{log-fold change})$: the FDR-adjusted $p$-value is the `adj.P.Val` output from `limma`, which is computed using the Benjamini-Hochberg procedure from the original $p$-values.

In addition, we considered multiple strategies of binarizing the significance of change in gene expression to cast the task as a classification problem. We found that $-\log_{10}(p\text{-value}) \times \text{sign}(\text{log-fold change})$ performs better as a perturbation representation according to the cross-donor retrieval (Figure 5).

## D  Limitations of Differential Expression Analysis

Predicting transcriptional differential effects using standard tools in scRNA-seq data from heterogeneous cell populations, such as PBMCs treated with targeted drugs, presents several challenges. Statistically, these tools must contend with batch effects, which can arise from processing times, reagents, or sequencing runs. Although adjustments for batch effects can be incorporated into the analysis design, the confounding of batch and treatment effects can still obscure true biological signals. Small sample sizes or high biological variability can further hinder accurate dispersion estimates of parametric methods like negative binomial models, thereby reducing statistical power. This limitation is especially pronounced in low-abundance cell types, where variability is high, and transcript detection is low. Additionally, high-quality, consistent data across all samples is recommended, which is challenging in practice. Insufficient sequencing depth and biological variability between donors can obscure true differential effects. Biologically, the complexity of PBMCs introduces further limitations. These cells engage in intricate interactions and signaling pathways that influence transcriptional

responses indirectly, complicating the identification of direct drug effects. Heterogeneity within PBMC populations and baseline variability among donors can obscure drug-induced transcriptional changes. To address these issues, experiments should use matched samples from the same donors and apply robust normalization methods. Additionally, differential gene expression analysis may miss regulatory effects at other levels, such as protein activity and epigenetic modifications. Complementing scRNA-seq data with other omics data, such as proteomics or epigenomics, and integrating these datasets can provide a more comprehensive view of drug effects.

# E   Summary of methods

Below, we describe 6 methods submitted by challenge participants and the control methods. Note that the methods needed to be updated to generalize to different datasets, which might have impacted their performance. Despite contacting the authors and our efforts, we suspect that the implementations of LSTM-GRU-CNN Ensemble and Transformer ensemble might have worsened their predictions. All of the methods were released under MIT license `https://www.kaggle.com/competitions/open-problems-single-cell-perturbations/rules`.

## E.1   Leaderboard winners

### E.1.1   LSTM-GRU-CNN Ensemble

Kaggle user jeannkouagou had the highest score on the private test with a method that integrated additional biological knowledge into the feature space. Towards that, they utilized ChemBERTa [30] embeddings for SMILES encodings of small molecules which resulted in notable improvements in predictive performance. Furthermore, a 5-fold cross-validation setting was utilized, incorporating three model architectures (LSTM [50], GRU [51], and 1d-CNN [52]) with multiple loss functions and three distinct input feature representations (initial, light, and heavy) to optimize model accuracy. The method also included additional data augmentation techniques, such as randomly replacing input features with zeros to simulate biological noise.

### E.1.2   Transformer ensemble

Kaggle user Elior Kalfon proposed a method based on a transformer [53] ensemble and scored 2nd place on the leaderboard. This method employed an ensemble of four transformer models, each with different weights and trained on slightly varying feature sets. Their method considered both the strategies for both feature normalization and data sampling. The feature engineering process involved one-hot encoding of categorical labels, target encoding using mean and standard deviation, and enriching the feature set with the standard deviation of target variables. Their method also considered to normalize data based on both mean value and standard deviation (std), or only mean value. A sophisticated sampling strategy based on K-Means clustering was employed to partition the data into training and validation sets, ensuring a representative distribution. The model architecture leveraged sparse and dense feature encoding, along with a transformer for effective learning.

### E.1.3   NN retraining with pseudolabels

Kaggle user Okon2000 scored 3rd place in the competition leaderboard using their multi-stage MLP approach. Both stages use an ensemble of MLPs that underwent individual hyperparameter optimization to select model dimensions, learning rate and dropout. The first round trains an ensemble of 7 MLPs to predict pseudolabels [31] for the entire test set. These pseudo labels are added to the training dataset and used in the second round, where an ensemble of 20 MLPs to predict the output. 4-fold cross-validation, averaged over 2 repeats per fold, was used to avoid overfitting. The submission finds benefit to replacing one-hot encoding with an embedding layer, but did not find improvements with various dataset denoising and label normalization schemes. The robustness of the model to increasing dataset size, noisy labels, and noisy inputs is examined, demonstrating small benefits to adding noise to training labels.

### E.2 Judge prize winners

### E.2.1 JN-AP-OP2

The solution by Antoine Passemiers and Jalil Nourisa earned the 1st judge prize. They employed a deep neural network architecture for perturbation modeling. Initially, the training data was encoded using a leave-one-out encoder based on unique pairs of compounds and cell types, converting the data into a format of (n_samples, n_genes, n_encode), referred to as X, where n_encode is 2. Then, the encoded data, X, was fed into the first multi-layer perceptron (MLP1). MLP1 processed X in a sample-wise manner and utilized fully connected layers to learn inter-gene relationships by sharing the encoded data across genes. Next, the output of MLP1 was concatenated with the original encoded data X to form a new representation of (n_samples, n_genes, 2*n_encode), which merged the learned encoding with the original encoding. This combined data was then inputted into a second multi-layer perceptron (MLP2) in a gene-wise manner, resulting in a final representation of n_samples * n_genes.

### E.2.2 ScAPE

Kagle user Los Rodríguez proposed their method named ScAPE, which won 2nd place for the Judge's award in the competition. With a similar design of chemCPA [13], the core of ScAPE is an auto-encoder that utilizes drug and cell features and outputs signed log(p-values). Specifically, it has separate encoders to learn the latent representations of cells and drugs, respectively. with noise introduced. ScAPE computes the features as the median of signed log(p-values) from differential expression analysis results calculated on single-cell level. In addition, it computes differential expression on pseudobulk level to get mean log(fold-changes) as extra information. The method uses cell features both in the encoding part and the decoding part of the neural network, which is non-probabilistic, as the authors didn't observe further advantages, either with respect to accuracy or generalization ability, with additional variational inference. Using cell latent features during decoding gives the method better scores in the leaderboard, though there's not much improvement observed during training. The model also employs a leave-one-drug-out cross-validation strategy to assess generalization to unseen drugs, which ensures robust predictions by leveraging both raw fold changes and the most variable genes, thus it results in a competitive performance. Besides, the authors also proposed several other designs of methods and benchmarked the performances. Their exploration of both the problem and methodology are well documented which could provide useful insights for further studies.

### E.2.3 Py-boost

This solution earned the third judge prize. Kaggle user AmbrosM implemented a gradient-boosted decision tree model using the py-boost framework [54]. The data is preprocessed in two ways before model training. First, -log10(pvalue)sign(lfc) values are converted to t-statistic. This mapping is continuous and bijective, so there is no loss of information. Second, the training data is compressed down to 50 dimensions with PCA. After training, model outputs are mapped through the (pseudo-)inverse of this PCA transform, then converted back into -log10(P-value)sign(lfc).

### E.2.4 Control methods

We implemented six control methods described below:

1. **Ground truth** (id: ground_truth): Return the test set as output.

2. **Constant zero** (id: zeros): Predict no differential expression for any of the samples.

3. **Random sample** (id: sample): Randomly sample counts from the training set per gene.

4. **Mean outcome**: We used three average-based baselines. One that averages over all of the compounds and cell types $\hat{y}_{ij} = \sum_{i=1}^{R} y_{\text{train}_{ij}}$ (id: mean_outcome), one that averages across all of the cell types for a given compound (id: mean_across_compound), and one that averages across all of the compounds for a given cell type (id: mean_across_celltypes).

Table 3: Comparison of coefficient of variation across Kaggle competition cell types. We observe high variation in T cells CD8+ and T regulatory cells in control compounds.

| Cell type | Dabrafenib | Belinostat | Dimethyl Sulfoxide |
|---|---|---|---|
| B cells | 0.319051 | 0.338520 | 0.307461 |
| Myeloid cells | 0.184550 | 0.275649 | 0.185540 |
| NK cells | 0.240455 | 0.577534 | 0.222283 |
| T cells CD4+ | 0.129801 | 0.162064 | 0.106406 |
| T cells CD8+ | 0.488251 | 2.288442 | 0.498569 |
| T regulatory cells | 0.411224 | 1.894598 | 0.317219 |

## F    Competition learnings

### F.1    Participant survey

We surveyed 35 competitors to learn more about the participants' backgrounds and their experience of the competition. 57% of respondents haven't worked with single-cell data before, and the same number never participated in a Kaggle competition before. 91% have not participated in an Open Problems competition before. The respondents come from 16 different countries. 31% work in industry, and 54% in academia. Only 9% used other single-cell datasets, and 3% used external references (e.g. KEGG or Gene Ontology) in their solutions.

### F.2    Outlier compounds

One of the 20 clusters identified by the Leiden algorithm (Appendix A.8) could not be conclusively labeled as belonging to any particular cell type. Over 96% of the cells in this cluster were from the wells of three compounds (Delanzomib, Oprozomib, and MLN2238), all of which shared the same mechanism of action, proteasome inhibition. To avoid biasing the perturbation prediction models with low-confidence cell type labels, these three compounds were removed from the dataset. Due to either low counts induced by toxicity or high variability in cell type proportions across replicates, three other compounds were also dropped: CGP60474, BXU45ZH6LI, and Alvocidib.

## G    Single-cell perturbation prediction evaluation

Single-cell perturbation models can also be applied to our benchmark task. According to a recent single-cell perturbation benchmark, PerturBench, a latent additive model performs best in this category [55]. We used the parameters from the PerturBench run that performed best on the sci-Plex dataset [17]. We trained the model on unnormalized counts. We then used the `limma` package for differential expression analysis on the predicted counts, and the resulting outcomes were used as model predictions. The latent additive model performed worse than our benchmark control methods according to mean row-wise RMSE and mean row-wise MAE (Table 4).

## H    Data analysis reproducibility

The code for reproducing the figures and data analysis, including cell type annotation and filtering, is available at `github.com/theislab/task-dge-perturbation-prediction-analysis`. The code is provided under MIT license.

## I    Benchmark details

Benchmark code is available at `github.com/openproblems-bio/task_perturbation_prediction`, DOI:10.5281/zenodo.11537124. The code of the benchmark is provided under MIT license.

## Differential Expression (DE) analysis
Calculating -log10(p-values)

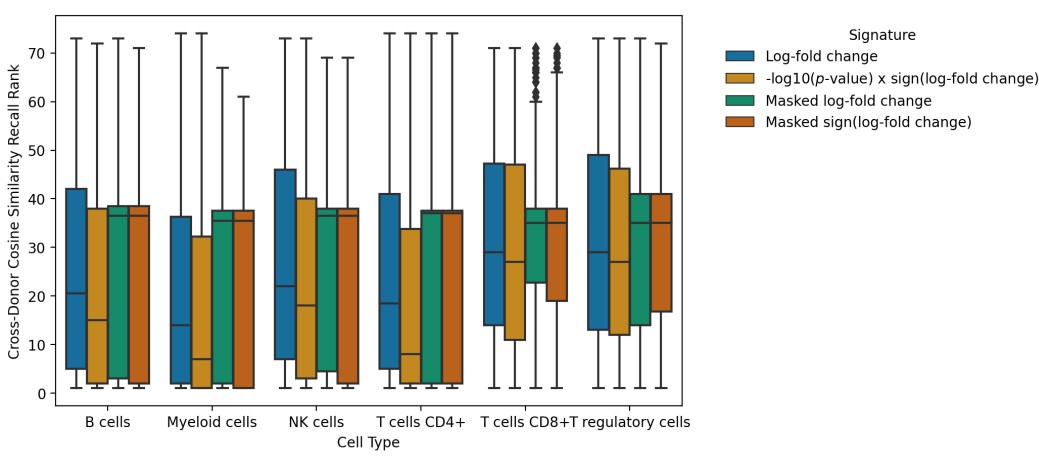

Donor 1    Donor 2    Donor 3

For training, the DE model is fit once to all samples from the training set of (compound, cell type) combinations.

For public and private test, the DE model is fit to all data. Contestants will not have access to this data.

This maintains privacy of the test set.

**Limma**

*Fitting a linear model with Limma*

$$f(g_j) = x_0 + x_1 c_i + x_2 l_i + x_3 d_i + x_4 p_i$$

Where $g_j$ is an indicator for each gene, and $c_i, l_i, d_i, p_i, t_i$ are indicators of the compound, library, donor, and plate of observation $i$. Here each observation is a pseudobulked gene expression profile for all cells of a given type from a given well in the experiment.

P-values are calculated based on the significance of model weights for each compound for each gene

Figure 6: High-level overview of the Kaggle competition dataset DE computation, including the design matrix.

Signature
- Log-fold change
- -log10($p$-value) x sign(log-fold change)
- Masked log-fold change
- Masked sign(log-fold change)

Figure 7: Cross-donor retrieval on the Kaggle competition dataset with cosine-similarity as a metric. The scores of T cells CD8+ and T regulatory cells stand out.

Table 4: Latent additive model comparison to OP3 benchmark models, sorted by mean row-wise RMSE.

| Model | Mean rowwise RMSE | Mean rowwise MAE |
|---|---|---|
| Ground truth | 0.0000 | 0.0000 |
| NN retraining with pseudolabels | 0.7562 | 0.5464 |
| LSTM-GRU-CNN Ensemble | 0.7921 | 0.5756 |
| Py-boost | 0.7957 | 0.5609 |
| Mean per cell type and gene | 0.8925 | 0.6437 |
| JN-AP-OP2 | 0.8965 | 0.6518 |
| Mean per gene | 0.8992 | 0.6356 |
| Zeros | 0.9179 | 0.6351 |
| Mean per compound and gene | 0.9428 | 0.6979 |
| Latent additive | 1.162 | 0.8223 |

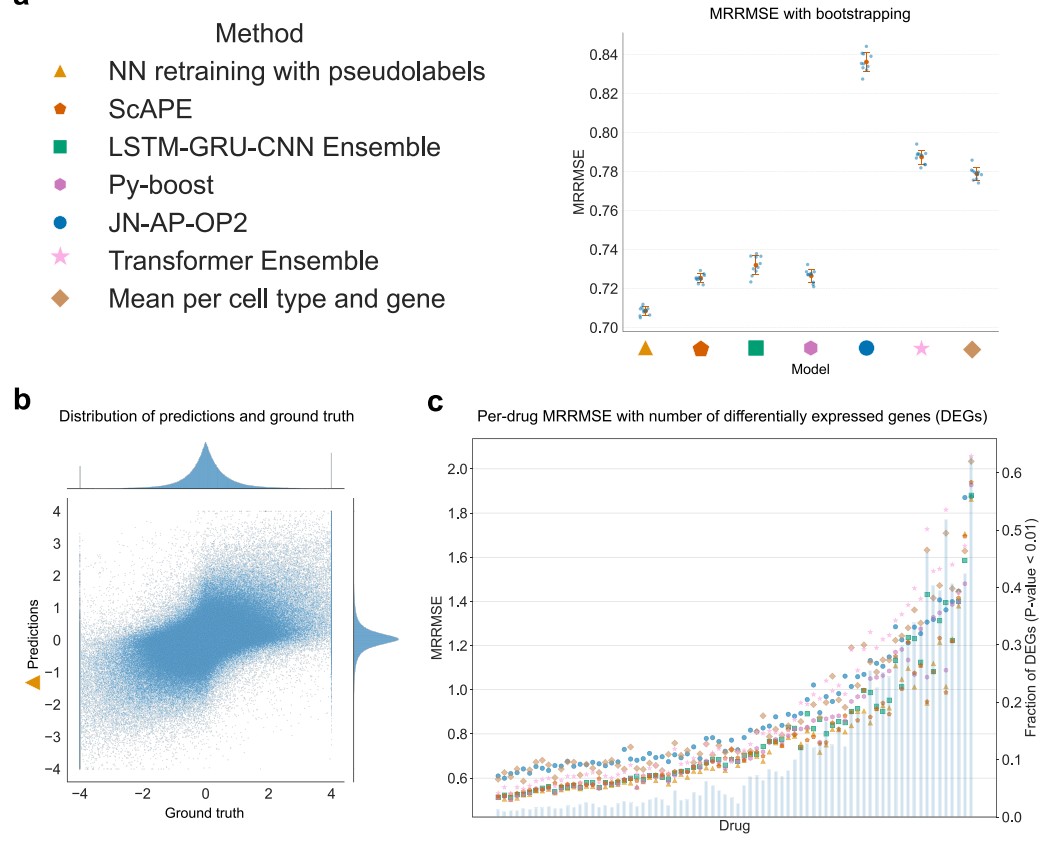

Figure 8: **Benchmark results. (a)** Results of rerunning the methods with dataset bootstrapping with 10 bootstraps. The error bars are standard deviation. Note that bootstrapping was performed by sampling cells in both the training and test sets. **(b)** Distribution of ground truth and the predictions of the top-performing method, NN retraining with pseudolabels. We note that the predictions are biased toward lower than true significance. **(c)** Per-drug MRRMSE and the fraction of genes for a given compound with a P-value lower than 0.01 (the latter shown with a bar chart). We note that the errors are larger for compounds with a high fraction of DEGs. The differences in errors across the methods and the baseline are smaller in samples with a low fraction of DEGs.

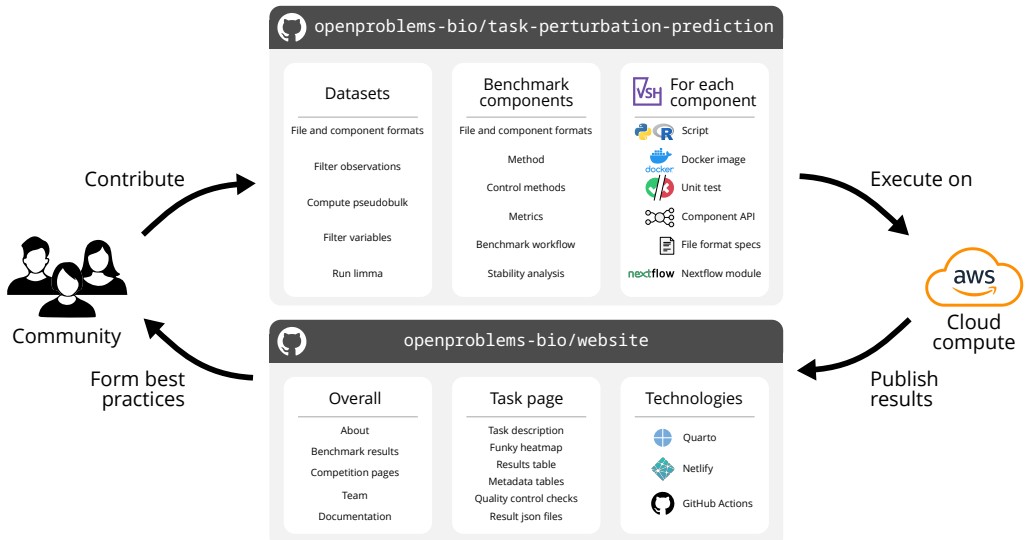

Figure 9: An overview of the technology stack of the perturbation prediction living benchmark within the OpenProblems ecosystem.

**Data formats**: To ensure interoperability between components, the repository uses AnnData [56] as the standard data format for both input and output files of components, and strict requirements are imposed on the format of these files.

**Components**: Workflows are comprised of Viash components and are themselves also Viash components [57]. A Viash component is a small amount of metadata combined with a script implemented in Python, R, Bash, or Nextflow. Viash can use this information to build a component-specific Docker container, and turn the component into a Docker-backed Nextflow workflow. These Nextflow workflows can be used as a standalone module, or as a submodule for another workflow.

### I.1 Workflows

The repository consists of three main workflows: `process_dataset`, `run_benchmark`, and `run_stability_analysis` (Figure 10).

### I.2 Workflow: Process dataset

The data processing steps used to transform the single-cell RNA-seq expression matrix into the Perturbation Differential Gene Expression (DGE) matrix for the perturbation prediction task (Figure 10 top). It consists of the following components:

- **Filter obs**: Remove low-quality observations from the dataset. The conditions are designed to exclude cells that could introduce bias or noise into the downstream analysis, such as cells from certain donors, cells treated with certain molecules, or certain cell types.
- **Compute pseudobulk**: Aggregate cell types into pseudobulks.
- **Filter vars**: Subset the genes
- **Limma on train**: Run limma on the train and control splits, per cell type and per small molecule. The resulting information is stored as an AnnData object we call DE train.
- **Limma on train and test**: Run limma on train, control and test split, per cell type and per small molecule. The resulting information is stored as an AnnData object we call DE test.
- **Extract ID map**: Extract a data frame containing a combination of the cell types and small molecules which methods will need to predict. The resulting information is called ID map.

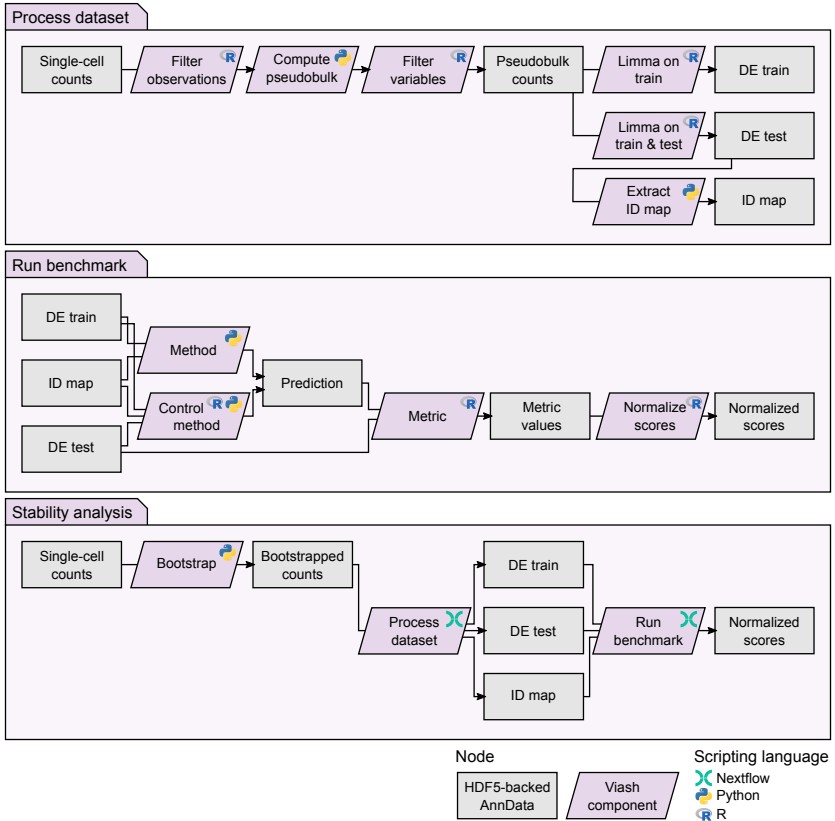

Figure 10: The different workflows used to perform the analyses in this study, `process_dataset`, `run_benchmark`, and `run_stability_analysis`. Each workflow uses HDF5-backed AnnData (h5ad) files (grey rectangle) as a common data format, and is comprised of Viash components (purple rhombus) implemented in Nextflow, Python, or R. Since each workflow is also a Viash component, it can in turn be used as a subworkflow of a larger workflow.

## I.3 Workflow: Run benchmark

Evaluate the performance of methods and control methods using a set of metrics (Figure 10 middle). This workflow accepts the DE train, DE test and ID map objects and inputs and runs the various control methods and methods on it. Each prediction generated by the methods is evaluated using each of the metrics. In the end, all output results is stored, alongside the dataset metadata, method metadata, metric metadata, and runtime resource information. The workflow consists of the following components:

- **Method**: A method for predicting the perturbation response of small molecules on certain cell types.

- **Control method**: A control method to serve as a quality control for the perturbation prediction benchmark.

- **Metric**: A metric to compare a perturbation prediction to the ground truth.

- **Normalize scores**: Normalise the metric values by min-max scaling the values between the worst control method result and the best control method result.

### I.4 Workflow: Stability analysis

This workflow is used to perform a stability analysis of the methods (Figure 10 bottom). It bootstraps the original single-cell counts matrix, and runs the Process dataset and Run benchmark workflows to perform a benchmark on each of the bootstrapped datasets. It consists of the following components:

- **Bootstrap**: This component bootstraps the single-cell dataset by sampling the same number of cells with replacement from the dataset.
- **Process dataset**: The process dataset workflow mentioned earlier.
- **Run benchmark**: The run benchmark workflow mentioned earlier.

### I.5 Execution environment

Workflows were executed on AWS Batch, where components could run completely in parallel depending on the topology of the workflow. Components were run on different instance types depending on the specific memory / CPU / GPU requirements of the component. The following is a list of suitable instance types depending on the requirements of the component:

- GPU required: `g4dn.8xlarge`, 32 vCPUs, 128 GB memory, 1 Nvidia T4 GPU.
- Low memory: `m4.2xlarge`, 8 vCPUs, 32 GB memory
- Medium memory: `m4.4xlarge`, 16 vCPUs, 64 GB memory
- High memory: `m4.10xlarge`, 40 vCPUs, 160 GB memory

All method components required a GPU to run, whereas dataset processing components, control methods, and metrics did not require a GPU to run.

A run of the `run_benchmark` workflow requires:

- 216 jobs on non-GPU and GPU instances
- Wall time: 3h 2m
- CPU time: 173 CPU hours
- Total memory: 232 GB
- Disk read: 24 GB
- Disk write: 27 GB

A run of the `run_stability_analysis` requires the following resources:

- 1271 jobs on non-GPU and GPU instances
- Wall time: 6h 52m
- CPU time: 2162 CPU hours
- Total memory: 3101 GB
- Disk read: 321 GB
- Disk write: 440 GB

## J   Informed consent for PBMC donors

For this study, we purchase commercially available human tissue samples from AllCells, Inc. AllCells is a tissue bank licensed by the State of California Department of Public Health, USA (Tissue Bank ID#: CTB 00080812). AllCells is responsible for maintaining IRB approval for all human subjects research. Below is one of the informed consent documents signed by one of the donors (name and signature redacted). More information is available from AllCells upon request.

1419

*Discovery Life Sciences, LLC*
*DLS-BB018-V.11*

| RESEARCH SUBJECT INFORMATION, CONSENT, AND AUTHORIZATION FORM | |
|---|---|
| TITLE: | Collection of Human Apheresis Specimens from Healthy Donors for Future Scientific and Medical Research |

**This consent form contains important information to help you decide whether to take part in a research study.**

> The study staff will explain this study to you. Ask questions about anything that is not clear at any time. You may take home an unsigned copy of this consent form to think about and discuss with family or friends.

➢ **Being in a study is voluntary – your choice.**
➢ **If you join this study, you can still stop at any time.**
➢ **Do not join this study unless all your questions are answered.**

## After reading and discussing the information in this consent form you should know:

- Why this study is being done;
- What will happen during the study;
- Any possible benefits to you;
- The possible risks to you;
- Other options you could choose instead of being in this study;
- How your personal health information will be treated, used, and **disclosed** during the study and after the study is over;
- Whether being in this study could involve any cost to you; and
- What to do if you have problems or questions about this study.

### Please read this consent form carefully.

1420

## RESEARCH SUBJECT INFORMATION, CONSENT, AND AUTHORIZATION FORM

**TITLE:**               Collection of Human Apheresis Specimens from Healthy Donors for Future Scientific and Medical Research

**PROTOCOL NO.:**    DLS-BB018-V.11
IRB Protocol #20130996

**SPONSOR:**       Discovery Life Sciences, LLC

**INVESTIGATOR:**   Timothy M. Howard, MD
800 Hudson Way
Huntsville, Al 35806
USA

**SITE(S):**

| | |
|---|---|
| Discovery Life Sciences, LLC
800 Hudson Way
Huntsville, Al 35806
USA | American Red Cross
100 Peartree Lane
Raleigh, NC 27600
USA |
| American Red Cross
1101 Washington St NW
Huntsville, Al 35801
USA | American Red Cross
2425 Park Road
Charlotte, NC 28203
USA |
| American Red Cross
700 Caldwell Trace
Birmingham, Al 35242
USA | American Red Cross
2751 Bull Street
Columbia, SC 29230
USA |
| American Red Cross
2179 Roswell Road
Marietta, GA 30062
USA | American Red Cross
100 Rustcraft Road
Dedham, MA 02026
USA |
| American Red Cross
337 Stoneridge Lane
Gahanna, OH 43230 | American Red Cross
7539 Oswego Road
Liverpool, NY 13090 |

**STUDY-RELATED**
**PHONE NUMBER(S):**    Discovery Life Sciences, LLC
Study Coordinator
256-327-9828 (24 Hours)

1421

## SUMMARY

We invite you to take part in a research study (the "Study"). The purpose of this consent and authorization form (the "Consent") is to help you decide if you want to be in the Study, and if you agree to have your health information used and disclosed for the Study. This Consent may contain words that you do not understand. Please ask the Study staff to explain any words or information that you do not clearly understand. You may have this Consent read to you.

Things to know before deciding to take part in a research study:

- The main goal of a research study is to learn things to help patients in the future.
- The main goal of regular medical care is to help each patient.
- Basic health information will be collected during the time you are taking part in this Study. This health information may be looked at and/or copied by Discovery, government agencies, and/or other groups associated with the Study.

If you take part in this research study, you will be given a signed copy of this Consent.

## PURPOSE OF THE STUDY

You are being invited to take part in a Study because human apheresis samples (for example, white blood cells and plasma) are needed to support research. Research on samples and health information can help scientists discover more about what causes diseases, how to prevent them, and how to cure them. The Study specifications will be provided to the subject by the Study Doctor.

Apheresis is the process of collecting particular parts of the blood (e.g. white blood cells, plasma, platelets) by passing the blood through an apheresis machine. The machine separates the part of the blood, collects those that are necessary, then returns the remainder of the blood back to the donor.

## DURATION OF THE STUDY

If you decide to take part in the Study, your participation is expected to last indefinitely or until you choose to no longer take part. There is no limit to the number of donors enrolled in the Study. The total number of donors expected to take part is unknown. Only adult donors will be included in this Study.

## PROCEDURES

If you decide to take part in the Study, after you sign this Consent, you will be required to complete a general health interview and meet the specified inclusion criteria to be eligible for the apheresis procedure.

### General Health Interview

Your vital signs (height, weight, blood pressure, pulse, and temperature) will be taken and recorded. Blood will be drawn and tested for blood counts (complete blood count and retic), metabolic function (comprehensive metabolic panel and hemoglobin A1C), lipid measurements (lipid panel), blood type (ABO/Rh), blood antibodies via direct and indirect antiglobulin tests, pregnancy, HLA typing, and the following list of diseases, as applicable: Covid-19, human immunodeficiency virus (HIV), hepatitis C virus (HCV), hepatitis B virus (HBV), hepatitis A

1422

virus (HAV), herpes simplex virus 1 (HSV-1), herpes simplex virus 2 (HSV-2), Varicella-zoster virus (VZV), Epstein-barr virus (EBV), cytomegalovirus (CMV), human herpesvirus 6 (variants A & B), human herpesvirus 7, Kaposi's sarcoma virus, West Nile virus (WNV), relevant cell-associated communicable disease agents and diseases (including human t-cell lymphotropic virus (HTLV)), human transmissible spongiform encephalopathy (including Creutzfeldt-Jakob disease), *Trypanosoma cruzi*, *Treponema pallidum*, syphilis, malaria, Zika Virus, and Parvovirus. Exclusive of the list above, any additional testing that is necessary to satisfy the project-specific inclusion and exclusion criteria may also be performed as long as the test does not require reporting to federal or state agencies. Women of childbearing potential will be tested for pregnancy. Pregnancy testing may be performed on blood, or a urine sample may be requested.

- If you test negative for all tested diseases and pregnancy, as applicable, and you meet the inclusion criteria as described below, you will be eligible for an apheresis procedure.

- If you have a positive test result for any tested disease, you will be removed from enrollment. If you have a positive pregnancy test, as applicable, and/or do not meet the inclusion criteria as described below, you will be enrolled in the Study but will not be eligible for an apheresis procedure at this time.

- If any of your disease tests are positive, the Sponsor will follow all applicable laws in the notification to appropriate agencies. The Sponsor will notify you of the positive result and request you seek follow up care with your general practitioner.

**Apheresis Procedures**

If you are eligible for an apheresis procedure you will be scheduled for the procedure within 3 weeks of the general health interview. Your vital signs (blood pressure, temperature, pulse) will be taken before the apheresis procedure begins. You will have an intravenous (IV) line (either a needle or catheter) placed in both arms. A nurse will monitor you and your vital signs will continue to be taken throughout the procedure. The apheresis procedure will take approximately four to five hours.

Once enrolled you are eligible to donate no more than one procedure of apheresis collection every sixty days (60) as long as you meet the inclusion/exclusion criteria described below. Mononuclear cells and other apheresis samples may be collected for a total product volume of no more than 550 mLs (a little less than 2 ½ cups).

The samples may be kept by the Sponsor in a bank and stored indefinitely.

In addition to the qualification and apheresis procedures, donor information about you (for example, age, race, and gender) and your pertinent medical information will be obtained from you or your donor record. This information will be linked to your samples. However, before the samples and information are released to any researcher, they will be given a special code without your name or private information on them that directly identifies you. The Site, Sponsor, Study Doctor, and Study staff may have access to the key that links this special code to your private information. However, no researchers will have access to your directly identifiable private information through this Study.

This Consent allows for more than one collection during your participation.

1423

The following procedures may be performed in this Study:

- Apheresis collection – Blood component separation procedure in which whole blood is removed from your vein and passed through a device that separates the blood into components. Particular components are collected, and the remaining components are returned back to you. Up to four blood volumes can be collected once every sixty (60) days.

- Nasal swab(s) collection – A procedure in which a sample of nasal secretions is taken. This is usually performed by wiping the inner nostril with a cotton-tipped swab.

- Nasopharyngeal swab(s) collection – A procedure in which nasal secretions from the back of the nose and throat is taken. This is usually performed by inserting a cotton-tipped swab into the nostril and rotating over the surface of the posterior nasopharynx.

- Urine collection – A procedure in which urine is collected in a sterile, plastic container.

- Venipuncture – A procedure in which blood is removed from one of your veins using a needle

## INCLUSION CRITERIA

- Age 18-70 years old (must be a legal adult in state of the Site)
- Weigh at least 110 lbs
- Baseline Blood Pressure: Systolic: 90 -180 mm Hg, Diastolic: 50-100 mm Hg
- Temperature: less than 99.5°F
- Pulse rate: 50-110 beats/minute and regular
- Negative for all tested diseases as listed in the General Health Interview section
- Hemoglobin:
    - Females: no less than 11.5 g/dcL
    - Males: no less than 12.2 g/dcL
- Hematocrit:
    - Females: No less than 35.2%
    - Males: No less than 38.2%

## EXCLUSION CRITERIA

- Donors who do not give informed consent
- Donors who do not understand the informed consent
- Women who are pregnant or breastfeeding
- Donors with any history of heart, lung, liver, or kidney disease
- Donors with any history of blood or bleeding disorders, including sickle cell disease
- Donors with any history of neurologic disorders
- Donors with any history of cancer
- Donors with any history of diabetes
- Donors with a positive test result for any disease tested for as listed in the General Health Interview section
- Steroid use within two weeks of apheresis procedure

1424

## RISKS AND DISCOMFORTS

There are potential unforeseen risks with any procedure. The known potential risks are as follows:

- Apheresis – potential risks include:
  - Citrate toxicity: muscle cramping, numbness, chills, tingling sensations. Citrate toxicities are managed symptomatically using oral calcium supplements.
  - Bleeding, bruising, irritation, infiltration, inflammation at the venipuncture sites, or risk of arterial puncture
  - Allergic reaction
  - Vasovagal episode: lightheadedness, hot flashes, nausea, vomiting, decreased heart rate, and decreased blood pressure.
  - Syncope: fainting, risk of injury/fall
  - Hyperventilation
  - Infection at venipuncture site
  - Air embolus from machine malfunction: gas bubble enters the blood stream
  - Long term effects of donor apheresis are unknown

- Nasopharyngeal swab(s) collection – potential discomfort or pressure is associated with this procedure.

- Venipuncture - potential risks include pain, bruising, lightheadedness, or, on rare occasions, infection.

- There are no known risks associated with nasal swab and urine collections. However, there may be infectious pathogens that can be spread to others. Hands should be washed thoroughly with antibacterial soap after collection of these biospecimens. There may be minor bleeding, bruising, or discomfort from the nasal swab.

- Confidentiality – There is a possible loss of confidentiality of your health information, although all reasonable efforts will be made to protect your information as described in this Consent.

Due to scientific advances or human error, your identity and health information may become known. Since DNA (the chemical that makes up genes) information is unique to you, in the future this link could occur. For this link to occur, it would require someone to take another sample from you, analyze the DNA, and compare it with the data resulting from this research project.

## COMPENSATION FOR INJURY

If you are injured, you should obtain treatment as you would for any other injury, or you may contact your Apheresis Nurse/Study Doctor who can refer you for treatment. There are no plans to compensate you for any injuries you suffer as a result of this Study.

1425

## USE OF SAMPLES AND INFORMATION

Samples may be used to explore possible links between different types of molecules (for example, DNA, RNA, proteins) and features of the people (for example, age, gender, family history of certain medical conditions). The medical conditions studied will be widespread including some that you may not have. None of the results will be linked directly to you. They will be linked only to the group of people. Researchers may perform a variety of tests including genetic tests, tests of the cells that make up your samples, DNA or RNA sequencing or gene editing and even future medical research that is currently unknown at this time.

Your samples may be stored in ways that allow the cells to grow and multiply. These multiplying cells may grow to what is called a cell line. Cell lines can be used for many future studies and these cells may be kept alive for many years. None of your donated samples will be infused into another human being.

Researchers may develop products based on things they learn from your samples. Any information obtained by the researchers as a result of testing your samples will not be provided to you, as applicable. Any applicable information provided to you will come from the Study Doctor. These researchers will use your samples as needed and destroy unused portions per government regulations. The tests done on your samples are for research purposes only.

## NEW INFORMATION

You will be told about any new information that might change your decision to be in this Study. You may be asked to sign a new Consent if this occurs.

## BENEFITS

If you agree to take part in this Study, there will be no direct medical benefit to you.

## COSTS

There are no costs to you for taking part in the Study.

## COMPENSATION FOR PARTICIPATION

You will be compensated for the time and effort you devote to this Study. The compensation for taking part is up to    .

The site where the procedure is performed will be reimbursed in accordance with separately negotiated agreements between the Site and the Sponsor.

## COMMERCIAL USES

Any samples you provide that are used in research may result in new products, tests, or discoveries. In some instances, these developments may have commercial value. There are no plans for you to share in any financial benefits from these products, tests, or discoveries.

1426

## ALTERNATIVE TREATMENT

The Study is for research purposes only. The only alternative is to not take part in this Study.

## VOLUNTARY PARTICIPATION AND WITHDRAWAL

Your participation in this study is voluntary. You may decide not to take part or you may leave the Study at any time. Your decision will not result in any penalty or loss of benefits to which you are otherwise entitled.

You may withdraw from taking part in the Study at any time. You do this by providing written or verbal notification to your Study Doctor or Study Staff. If you withdraw your permission, you will not be able to continue taking part in this Study. Upon withdrawal, information that has already been gathered and samples already distributed before the date of withdrawal may still be used to make the research reliable. Remaining samples collected during the period of time you had given consent may be used for research. Information that has already been gathered will be maintained to ensure the accuracy of the research.

Your participation in this Study may be stopped at any time by the Study Doctor or the Sponsor without your consent.

## SOURCE OF FUNDING FOR THE STUDY

Funding for this Study is provided by Discovery Life Sciences, LLC, the Sponsor.

## QUESTIONS

If you have questions, concerns, or complaints, or think this research has hurt you or made you sick, talk to the research team at the phone number listed above on the second page.

This research is being overseen by an Institutional Review Board ("IRB"). An IRB is a group of people who perform independent review of research studies. You may talk to them at 855-818-2289 or researchquestions@wcgirb.com if:

- You have questions, concerns, or complaints that are not being answered by the research team.
- You are not getting answers from the research team.
- You cannot reach the research team.
- You want to talk to someone else about the research.
- You have questions about your rights as a research subject

## AUTHORIZATION TO USE AND DISCLOSE INFORMATION FOR RESEARCH PURPOSES

Federal regulations give you certain rights related to your health information. These include the right to know who will be able to get the limited information and why they may be able to get it. The Study doctor must get your authorization (permission) to use or give out any health information that might identify you.

1427

**What information may be used and given to others?**

If you choose to be in this Study, the Study doctor will get limited personal information about you. This may include information that might identify you. The Study doctor may also get limited information about your health including:

- Past, present, and future medical records
- Research records
- Questionnaire information collected as part of the Study
- Records about your Study visits
- Disease registry information.

**Who may use and give out information about you?**

The limited information about your health may be used and given to others by the Study Doctor, Study staff, or the Sponsor. They might see the research information during and after the Study.

**Who will get this information?**

The Sponsor of this Study will have access to your limited personal and medical information. Sponsor means any persons or companies that are:

- working for or with the Sponsor, or
- owned by the Sponsor.

Researchers will receive certain limited information about you. This limited information will not directly identify you.

The limited information about you and your health, which might identify you, may be given to:

- The U.S. Food and Drug Administration (FDA),
- Department of Health and Human Services (DHHS) agencies,
- Governmental agencies in other countries, and
- Institutional Review Board (IRB).

**Why will this information be used and/or given to others?**

The Sponsor will analyze and evaluate the results of the Study. The Sponsor will be visiting the research site. They will follow how the Study is done, and they will be reviewing your limited information for this purpose.

The limited information about you may be given to researchers to carry out the Study, but your identity will not be disclosed.

The limited information about you may be given to the FDA. It may also be given to governmental agencies in other countries. The limited information may be used to meet the reporting requirements of governmental agencies.

The results of this research may be published in scientific journals or presented at medical meetings, but your identity will not be disclosed.

1428

**What if I decide not to give permission to use and give out my limited health information?**
Then you will not be able to be in this Study.

**May I review or copy the limited information obtained from me or created about me?**
Yes, but only after the Study is closed.

**May I withdraw or revoke (cancel) my permission?**
Yes, but this permission will not stop automatically.

You may withdraw or take away your permission to use and disclose your limited health information at any time. You do this by notifying the Study Doctor or Study staff in writing or verbally. If you withdraw your permission, you will not be able to stay in the Study.

When you withdraw your permission, no new health information identifying you will be gathered for the Study after that date. Once the Sponsor receives your withdrawal notice, it will not further disclose your limited information, but it may still use the limited information to make the Study reliable.

However, your withdrawal will not affect any action that has already been taken in reliance on your authorization. For example, if the Sponsor has already released your limited information to another researcher for future use, it may continue to be used and disclosed, and it will not be possible to get the limited information back.

**Is my limited health information protected after it has been given to others?**
The Sponsor has processes in place to protect your limited identifying information; for example, your name is replaced by a number and you are referenced only by that number with others who do not have the ability to tie that number back to your name. However, there is a risk that your limited information will be released to others who may not have the same legal obligation to protect that limited information.

**When does my permission to use my limited information expire?**
There is no current plan to end the Study. Your limited information may be held in a repository (or multiple repositories) indefinitely, and your permission to use this limited information will not expire.

1429

**CONSENT TO PARTICIPATE IN THE STUDY**

I have read this Consent (or it has been read to me). All my questions about the Study and my part in it have been answered. I freely and voluntarily consent to take part in this Study.

By signing this Consent, I give permission for my samples and limited health information to be used and stored for current and future research of my medical diagnosis or other medical diagnoses.

By signing this consent form, I have not given up any of my legal rights.

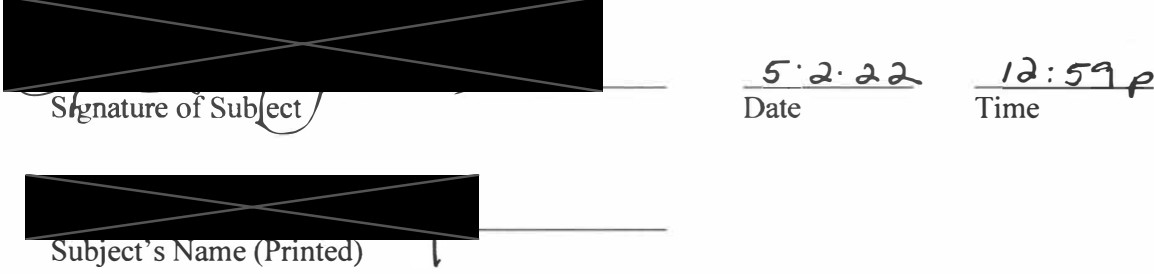

_______________________________     _5·2·22_     _12:59p_
Signature of Subject                      Date           Time

_______________________________
Subject's Name (Printed)

**PERSON CONDUCTING INFORMED CONSENT DISCUSSION:**

I confirm that the Study was thoroughly explained to the subject, including but not limited to the risks and benefits of participation, and that it is voluntary. I reviewed the Consent with the subject and answered the subject's questions. The subject appeared to have understood the information with verbal recall about the Study upon my questioning.

_Malicia George_______________     _5/2/22_     _12:59 PM_
Signature of Person Conducting the       Date           Time
Informed Consent Discussion

_Malicia George_______________
Printed Name of Person Conducting the
Informed Consent Discussion

1430

**-------------------------- Use this witness section only if applicable --------------------------**

If this Consent is read to the donor because the donor is unable to read the Consent, an impartial witness not affiliated with the research or investigator must be present for the consent and sign the following statement:

I confirm that the information in the Consent and any other written information was accurately explained to, and apparently understood by, the donor. The donor freely consented to be in the Study.

_______________________________       __________        __________
Signature of Impartial Witness              Date                Time

_______________________________
Printed Name of Impartial Witness

Note: This signature block cannot be used for translations into another language. A translated Consent is necessary for enrolling donors who do not speak the language of this consent.

