# OpenReview forum: "A benchmark for prediction of transcriptomic responses to chemical perturbations across cell types"
_NeurIPS.cc/2024/Datasets_and_Benchmarks_Track — NeurIPS 2024 Track Datasets and Benchmarks Spotlight_

### Official Review · Reviewer_daUw · 2024-07-21
**Transcriptomic dataset & related competition**

**Rating:** 10
**Confidence:** 3
**Correctness:** N/A
**Clarity:** N/A

**Review:**

Extremely well written and high quality submission, proposes a novel dataset for single cell gene expression, but also gives a detailed summary of a competition that was ran together with it. It's worth noting that a lot of detail is given to describing the thought process at every stage, both from the point of view of dataset desiderata, but also the competition itself and lessons learned during it. In a sense, it's a beautiful example of science happening in real time, done together with the community.

**Strengths:**

- Very clearly written.
- Justifies every step on the way.
- What I expect to be a highly impactful dataset.

**Additional Feedback:**

N/A

**Documentation:**

N/A

**Opportunities For Improvement:**

Create that consortium you mention and generate more data.

**Relation To Prior Work:**

N/A

**Summary And Contributions:**

A very well written paper, both proposing a novel dataset on single cell gene expression (across multiple cell lines and drugs) and documenting a competition that was ran on its earlier version.

---

> ### Author Rebuttal · Authors · 2024-08-16
>
> We appreciate the reviewer's time and supportive feedback on our manuscript. We find the reviewer's opinion of the high impact of the dataset, and the exceptional quality of the manuscript very encouraging. We fully agree that an effort to create a consortium for coordinated data generation is needed and we are eager to pursue it. If you have any new comments or questions, please let us know. We are committed to addressing them promptly.

---

### Official Review · Reviewer_Cdu1 · 2024-07-25
**A valuable high quality single cell perturbation response dataset**

**Rating:** 7
**Confidence:** 4
**Correctness:** Yes.
**Clarity:** Yes.

**Review:**

Pros:

- High-quality single-cell PBMC dataset with comprehensive perturbation screens
- Crowdsourced benchmark with over 1,000 teams provides solid methodological baselines for perturbation predictions

Cons:

- There is a lack of summary about what really works regarding model design and data processing from the top winning solutions. The "Lessons learned" is a good start, but I'm expecting something to help guide future study and method development for perturbational predictions in single-cell analysis.

Questions:

- I'm skeptical about the evaluation based on regressing the p-value as numerical objectives. The authors mentioned that they clipped the p-value to 1e-4 as the returned p-value from limma-voom for this dataset could be as low as 1e-180, proving that regressing the p-value numerically might be suboptimal. Why not just treat it as a classification problem that predicts whether a gene is differentially expressed (and its sign)? Is the exact numerical p-value returned from limma-voom a biologically significant value to predict?
- The perturbation task aims to predict perturbation effects on a subset of held-out cell types, which answers the following question: "Given the perturbational responses of a molecule to some cell types, how would it affect other cell types?" However, this has very limited significance. As pointed out in the intro, there are ~10^60 drug-like molecules. Wouldn't a better question here be: "Given the perturbational responses of some molecules, what would the perturbational response look like (on a particular cell type that we know) on a new molecule?"

**Strengths:**

See pros.

**Additional Feedback:**

See questions.

**Documentation:**

Some documentation and descriptions are available on the Kaggle competition page. More detailed descriptions are available in the submitted appendix.

**Ethics:**

No.

**Limitations:**

Yes.

**Opportunities For Improvement:**

- Line 148: "We determined that..." This sentence is quite confusing. What does the "distance between observations of the same compound" mean? Please try to find a clearer description.
- See cons.

**Relation To Prior Work:**

Yes.

**Summary And Contributions:**

This work presented a single-cell dataset with comprehensive perturbation screening across 146 compounds on the human blood cells. The authors took extreme care in data quality control and study design, making the resulting dataset a valuable asset to the computational single-cell analysis community. The dataset is coupled with a live benchmark for predicting perturbational outcomes, which was held on Kaggle with over 1,000 participating teams. The top-performing methods were briefly mentioned, with some discussion on key lessons learned.

---

> ### Author Rebuttal · Authors · 2024-08-16
>
> We thank reviewer Cdu1 for a detailed and thoughtful review of our manuscript. We appreciate your recognition of the strengths of our work, particularly the valuable drug-perturbation dataset with single-cell readouts and the crowdsourced benchmark that was hosted as a popular Kaggle competition. If any part of our response is unclear, or if you have additional comments or questions, please let us know. We are committed to addressing them promptly.
>
> Point-by-point response:
> >Cons (...)
>
> We are similarly interested in understanding what model architectures end up really working in the long term as we monitor our living benchmark seeded by models during our short three-month competition. We believe these models are the beginning of a field of models for cross-cell-type perturbation prediction, and at this stage, there are a wide variety of model architectures, training regimes, and ensembling methods among the top methods.
>
> We highlighted the key trends among the top-scoring methods in sec. 4.1 and 4.2 and provided detailed descriptions of these top methods in Ap. E. Beyond the top-performing models, the Kaggle page[1] contains > 250 notebooks with code for numerous other models submitted by the participants, along with detailed descriptions of their findings and solutions.
>
> We believe that the important problem of accurate modeling of the compound perturbations across cell types is far from being solved. For example, almost none of the winning methods included pre-training on external data, likely because developing such a model requires more time than allowed for in the competition itself. However, we expect such pre-training to yield even greater performance as seen in the recent explosion of self-supervised representation models for single-cell data.
>
> We are eager to draw a larger community to our benchmark and data to see long-term trends appear, which is a major motivation for publishing this work.
>
> >Questions: I'm skeptical about the evaluation (...)
>
> First, we note that log p-values are commonly used as a representation of differential expression for chemical perturbations[2,3]. However, the reviewer raises a good point, one which we explored when setting up the competition but did not include in this manuscript.
>
> Before launching the competition, we compared several different representations, aiming to find one that was the most consistent across donors while preserving the identifiability of perturbations from one another. Although we did not include them in the manuscript for brevity, we did consider multiple strategies of binarizing the data to create a classification problem.
>
> However, in this analysis, we found that $pert_{c, t, g} = -\log_{10}(p_{c, t, g}) \times sign(L_{c, t, g})$ (Eq1) is a better perturbation representation according to our cross-donor retrieval metric. We will expand this analysis in the camera ready to include the binarized classification representations in our comparison. We are also including the results in the attached pdf where "DE X" is binarized representation, with sign for direction, and X as the threshold.
>
> >Questions: The perturbation task (...)
>
> Although we can understand where the reviewer is coming from, we disagree that predicting perturbation effects in unseen cell types has limited significance. There are many axes of experimental design to consider when measuring the effect of a compound in a particular cell context, such as perturbation, cell type, timing of perturbation, timing of measurement, dose, cell culture condition, dosage, genetic background, etc. For a real-world drug discovery campaign, all of these factors are important.
>
> To illustrate the importance of predicting across cell types, consider how useful it would be to be able to predict the impact of a clinical candidate drug on every cell in the human body by only measuring that drug in a few cell types. This would make it possible to identify off-target effects in hepatic, cardiac, or nervous system tissue early and prevent toxicity issues before the drug goes into the clinic. See for example [4] on using perturbation signatures in hepatocytes to assess drug-induced liver injury.
>
> One of the reasons we decided to start with cross cell type prediction is that this problem is far more tractable than predicting across chemical space. The number of distinct cell types in the human body is estimated to be in the range of 200-1000[5]. In contrast, chemical space is far more vast, even if one restricts to the ~40B compounds in synthetically tractable virtual libraries. This means that a perturbation dataset that features a small number of cell types can still be a robust benchmark for the general task of cross-cell type prediction, while a dataset for predicting perturbation effects on unseen compounds would have to include orders of magnitude more compounds to be an equally robust benchmark. Nonetheless, we hope that continued increases in throughput for perturbation experiments will enable the construction of such a benchmarking dataset in the future.
>
> >Opportunities For Improvement: Line 148 (...)
>
> We agree with the reviewer that the sentence on cross-donor retrieval may be confusing. To address this, we have revised line 148 for clarity. Note that we refer to the appendix for a detailed and formulaic description of cross-donor retrieval.
>
> [Line 148] "We determined that an optimal representation of perturbation effect should allow perturbations of the same compound to be distinguished from perturbations of other compounds across donors. We propose
> cross-donor retrieval (Ap.C) as a method for quantifying these properties."
>
> [1] https://www.kaggle.com/competitions/open-problems-single-cell-perturbations/code
>
> [2] https://doi.org/10.1038/s41576-023-00586-w, Heumos et al. 2023
>
> [3] https://doi.org/10.1016/j.chembiol.2017.03.016, Drawnel et al. 2017
>
> [4] https://doi.org/10.1038/ncomms15932, Kohonen et al. 2017
>
> [5] https://doi.org/10.1073/pnas.2303077120, Hatton et al. 2023

---

> > ### Comment · Reviewer_Cdu1 · 2024-08-17
> > **Thanks for the rebuttal**
> >
> > I appreciate the authors' thorough responses. I'm satisfied with the explanations and efforts in deciding why not to binarize the p-values. I do not have any further questions.

---

### Official Review · Reviewer_p5Db · 2024-07-25
**Solid dataset, interesting metrics, and real-world models**

**Rating:** 7
**Confidence:** 4
**Clarity:** yes the paper is well written with gr…

**Review:**

This paper addresses an important problem in the field of chemical perturbation modeling: lack of data, and proper data splits and evaluation metrics. The work is significant as it ran one of the major public competitions on this topic. It is one of the few works it is original.

**Strengths:**

- Compiles and publishes a new dataset that is scientifically rigorous and provide splits that mirror biological applications by holding out entire perturbations on specific cell types
- A novel significance metric that captures replicates closeness across donors
- A list of publicly submitted models and their findings

**Additional Feedback:**

No additional feedback

**Correctness:**

yes the paper goes to great length to construct the dataset in a scientifically sound manner

**Documentation:**

yes

**Ethics:**

no ethical concerns

**Limitations:**

yes the authors were upfront about many limitations

**Opportunities For Improvement:**

I would be interested to see how models, other than those submitted to the competition, would fare.

**Relation To Prior Work:**

yes

**Summary And Contributions:**

This paper introduces the dataset used in the Kaggle' "open problems -single cell perturbations" competition. Overall, this paper discusses the following:
- Introduces the dataset and the science behind collecting this data and the creation of its test data; it pays special attention to replicates, diversity of perturbations, relevance to disease, and balances cellular heterogeneity. It further splits the test data to include perturbations for a whole cell line that have not been seen during training.
- Introduces a novel significance estimation based on a heuristic cross-donor retrival that uses the replicates data to ensure closeness of similar perturbation across donors. The resulting p-metric computed via limma captures both he direction and the statistical significance of the effect
-  Uses standard evaluation metrics such as cosine similarity
- Describes the competition, its leaderboard, the top submission and their insights

Overall the paper is well structured and provides valuable insights.

---

> ### Author Rebuttal · Authors · 2024-08-16
>
> We thank reviewer p5Db for the thoughtful assessment of our manuscript and their valuable suggestion. We are pleased to hear that the reviewer finds our paper to be an original and significant contribution to the field, in addition to being well-structured and insightful. If any part of our response is unclear, or if you have additional comments or questions, please let us know. We are committed to addressing them promptly.
>
> Point-by-point response:
> >I would be interested to see how models, other than those submitted to the competition, would fare.
>
> We are also keen to see how models other than those submitted to competition fare on the benchmark, which is why we created this living benchmark hosted at https://openproblems.bio/results/perturbation_prediction/. Indeed, one of the main motivations for publishing this work is to announce the creation of this benchmark and encourage ongoing community contribution.
>
> We note that while the manuscript focuses on the top-performing models from the competition, the Kaggle page[1] includes the code for numerous other models submitted by the participants in over 250 notebooks as well as detailed descriptions of their solutions. The solutions include models inspired by publications such as CPA[2].
>
> Nevertheless, to address this concern, we are currently working to add additional published methods including CPA[2] and biolord[3] to our benchmark, and will include them in the camera-ready. We note that adding more methods is time and resource-intensive. For example, training CPA with the suggested parameters with two NVIDIA A100-PCIE-40GB GPUs can take about 80 hours. Computing the 10 bootstraps needed for Fig. 7a will take over a week.
>
> [1] https://www.kaggle.com/competitions/open-problems-single-cell-perturbations/code
>
> [2] Predicting cellular responses to complex perturbations in high‐throughput screens, Lotfollahi et al. 2023
>
> [3] Disentanglement of single-cell data with biolord, Piran et al. 2024

---

### Author Rebuttal · Authors · 2024-08-16

# General response

We sincerely thank all the reviewers for their constructive feedback. As authors, we greatly appreciate the opportunity to improve the scientific quality of our work by clarifying statements, adding additional experimental results, and addressing the critiques provided.

Overall, all reviewers recognized the significance of the contributions introduced by our paper, including the dataset, competition, and benchmark, as well as the clarity of the manuscript. The reviewers also confirmed the correctness of the work and did not raise any ethical concerns or issues with the documentation.

To summarize, reviewer daUw did not raise any concerns. Reviewer p5Db proposed including more models in the benchmark, and reviewer Cdu1 asked several questions, which we have addressed in our corresponding response and updated in our manuscript.

We once again thank the reviewers for their valuable time and hope our responses address their concerns. We look forward to any further feedback and discussion.

Best regards,

The authors

---

### Decision · Program_Chairs · 2024-09-26

**Decision:**

Accept (Spotlight)

**Comment:**

This paper introduces a new dataset and benchmark for transcriptomic pertubations. The dataset and benchmarked models have been introduced and developed in a 2023 Neurips competition.

This is a strong original contribution. All reviewers really appreciate it and are in favour of acceptance.

It could be a good candidate for an oral presentation because there are a lot interesting findings of wide interest in this paper.